# Prioritized Federated Learning: Leveraging Non-Priority Clients for Targeted Model Improvement

**Aditya Narayan Ravi**                                                    *anravi2@illinois.edu*
*Electrical and Computer Engineering*
*University of Illinois Urbana-Champaign*

**Ilan Shomorony**                                                          *ilans@illinois.edu*
*Electrical and Computer Engineering*
*University of Illinois Urbana-Champaign*

**Reviewed on OpenReview:** *https://openreview.net/forum?id=FR8dvo6q8i*

## Abstract

Federated Learning (FL) is a distributed machine learning approach to learn models on decentralized heterogeneous data, without the need for clients to share their data. Many existing FL approaches assume that all clients have equal importance and construct a global objective based on all clients. We consider a version of FL we call Prioritized FL, where the goal is to learn a weighted mean objective of a subset of clients, designated as priority clients. An important question arises: How do we choose well-aligned non-priority clients to participate in the federation, while discarding misaligned clients? We present FedALIGN (Federated Adaptive Learning with Inclusion of Global Needs) to address this challenge. The algorithm employs a matching strategy that chooses non-priority clients based on how similar the model's loss is on their data compared to the global data, thereby ensuring the use of non-priority client gradients only when it is beneficial for priority clients. This approach ensures mutual benefits as non-priority clients are motivated to join when the model performs satisfactorily on their data, and priority clients can utilize their updates and computational resources when their goals align. We present a convergence analysis that quantifies the trade-off between client selection and speed of convergence. Our algorithm shows faster convergence and higher test accuracy than baselines for various synthetic and benchmark datasets.

## 1 Introduction

Federated Learning (FL) (McMahan et al., 2017; Kairouz et al., 2021) is a distributed learning setting where a set of clients, with highly heterogeneous data, limited computation, and communication capabilities, are connected to a central server. For $N$ clients, the goal of FL is for the central server to learn a model that minimizes

$$F(\mathbf{w}) := \sum_{k=1}^{N} p_k F_k(\mathbf{w}), \tag{1}$$

where $\mathbf{w} \in \mathbb{R}^m$ is the model parameter. The objective function $F_k(\mathbf{w})$ is the local objective function of the $k$th client, and $p_k$ is the fraction of data at the $k$th client. A key constraint of FL is that the server wishes to minimize $F(\mathbf{w})$ without directly observing the data at each of the $N$ clients.

A popular method to solve (1) in the FL setting is Federated Averaging (FedAvg) (Brendan McMahan et al., 2017). In FedAvg, $E$ iterations of Stochastic Gradient Descent (SGD) are performed locally at the clients before each communication round. After the $t$th local iteration, if $t$ corresponds to a communication round (i.e., $t \pmod E = 0$), a participating client $k$ sends the updated model parameter $\mathbf{w}_t^k$ to the server, which aggregates them as $\mathbf{w}_t = \sum p_k \mathbf{w}_t^k$, updates the global model, and shares it with all clients.

The data across the clients are generally heterogeneous, in which case the model updates from the different clients are not well aligned. In Li et al. (2020a), the authors use a heterogeneity measure $\Gamma$, given by

$$\Gamma := F^* - \sum_{k=1}^{N} p_k F_k^*, \tag{2}$$

which captures the gap between the global objective's minimum $F^*$ and a weighted sum of the minima $F_k^*$ of each local objective $F_k$. A convergence analysis for FedAvg (in the strongly convex case (Li et al., 2020a)) implies that the expected convergence error $E[F(\mathbf{w}_T)] - F^*$ at time step $T$ scales as

$$E[F(\mathbf{w}_T)] - F^* \sim (C + \Gamma)/T, \tag{3}$$

where $C$ is a constant. Intuitively, a more severe misalignment between the local and global objectives leads to a larger $\Gamma$ and to worse training performance. In particular, this suggests that FL systems that operate with a set of clients with reasonably aligned objectives may perform better.

**Prioritizing Clients in the Model.** The idea of selecting a subset of clients to include in each communication round in order to tackle heterogeneity is not new (Fu et al., 2022; Karimireddy et al., 2021; Li et al., 2018). In addition to reducing heterogeneity, client selection strategies have been proposed in the FL literature to accelerate convergence rates (Cho et al., 2020; Chen et al., 2020), promote fairness (Sultana et al., 2022), and bolster robustness (Nguyen et al., 2021). These methods emphasize specific clients during the aggregation process based on the task at hand, yet they cannot account for real-world applications with intrinsic prioritization.

Specifically, in various real-world applications, some clients are inherently prioritized, compared to others. In subscription-based services, for instance, it is often the case that some clients have a paid subscription, while others use a free version of the service. One example is online streaming services, which may provide perks such as ad-free content only to paid subscribers. In such cases, the central server may want to optimize a model (e.g., a recommendation system) that prioritizes the paid subscribers, while still taking advantage of the data provided by all clients. Another example are internet service providers, which have a set of paying clients, but may also provide free-access Wi-Fi hotspots. In such cases, the free service may be provided in exchange for some input from the clients, such as data or computational resources. These clients are likely to have highly heterogeneous data, and may not be willing to directly share their data, which fits naturally into a FL setting.

In these cases, the global objective function $F(\mathbf{w})$ should only take into consideration the priority clients, and it is important to understand whether the non-priority clients can be leveraged to improve the prioritized global objective. Is it possible to incentivize "well-aligned" non-priority clients to participate and contribute to model improvement, while discarding the updates from "misaligned" clients? To formalize this question, we propose the Prioritized Federated Learning (PFL) setting. We discuss additional motivating scenarios for PFL in Section 2.

**Proposed Model.** In PFL, there are $N$ clients, and a subset $\mathcal{P} \subset [1:N]$ is designated as *priority* clients. Similar to (1), the goal is to minimize $F(\mathbf{w}) = \sum_{k \in \mathcal{P}} p_k F_k(\mathbf{w})$. The remaining clients are designated as *non-priority* clients. Non-priority clients, if chosen wisely and incorporated into the learning process, can potentially accelerate the model's convergence through additional gradient steps. To facilitate this, we propose to devise an algorithm capable of discerning well-aligned non-priority clients and aggregating their updates along with those from the priority clients. The critical challenge here lies in finding the right balance: including more non-priority clients might expedite the convergence but could also introduce bias into the global model, especially if these clients are not well-aligned with the primary objective. Conversely, implementing more stringent selection criteria ensures only the most well-aligned clients contribute, but at the expense of faster convergence from additional updates. Against this backdrop, our paper will address two questions that arise in this setting: (1) What criterion should we use to effectively select well-aligned non-priority clients without compromising the integrity of our learning objective? (2) Can we establish a theoretical framework that quantifies the trade-off between accelerated convergence and potential bias introduced by including updates from non-priority clients?

**FedALIGN and Our Contributions.** To address the questions proposed for this PFL setting, we propose FedALIGN (Federated Adaptive Learning with Inclusion of Global Needs). FedALIGN's strategy for non-priority client inclusion can be combined with a variety of existing algorithms. In every communication

round, the server transmits not only the global model but also its associated loss to all participating clients. Priority clients invariably contribute to the global model's updates. Non-priority clients, on the other hand, participate only if their local loss is on par with or lower than the communicated global loss. Notice that this is a key step in incentivizing client participation, since well-aligned non-priority clients only participate if they get a satisfactory model. The server then incorporates only those updates from clients that are close to the global loss.

The core advantage of FedALIGN lies in its adaptive capacity. It exploits well-aligned non-priority clients when their contribution is beneficial, also incentivizing their participation. At the same time, it prevents potential harm to the model from misaligned clients, thereby protecting the interests of priority clients. Theoretical analyses suggest that FedALIGN effectively navigates data heterogeneity amongst priority clients, accelerating convergence at the expense of a manageable bias term. This bias can be subsequently removed in later communication rounds allowing FedALIGN to converge to the optimal solution.

We present the FedALIGN algorithm as a modification of FedAvg (although it can be built upon other algorithms too, as discussed in the supplementary material). To analyze its performance, we adapt the convergence analysis from Li et al. (2020a). For FedALIGN, the convergence error is given by:

$$E[F(\mathbf{w}_T)] - F^* \sim \frac{C + \theta_T \Gamma}{T} + \rho_T. \tag{4}$$

The factor $\theta_T \in [0, 1]$ captures the reduction in heterogeneity achieved by FedALIGN. The more non-priority clients are included in all communication rounds till $T$, the lower the value of $\theta_T$. This is balanced with a bias term $\rho_T$, which captures the misalignment in the selected non-priority clients.

The terms $\theta_T$ and $\rho_T$ exhibit an inverse relationship and depend on the desired proximity between global and local losses. Specifically when $\theta_T = 1$ and $\rho_T = 0$ (we show in Section 3 that this choice achieved by ignoring all non-priority clients), we recover (3), the convergence rate for FedAvg. The tradeoff between $\theta_T$ and $\rho_T$ enables us to fine-tune these terms, favoring a rapid convergence initially, and gradually adjusting the bias as needed. Indeed experiments implementing FedALIGN shows that exploiting this trade-off leads to faster convergence and better performance over baseline algorithms, both on synthetic data and many benchmark datasets (FMNIST, EMNIST and CIFAR10).

**Prioritization and Fairness in AI.** In our exploration of PFL, it is crucial to talk about fairness. At the heart of this discussion is the fact that PFL is developed to model real-world scenarios such as subscription-based services where a large fraction of the population (non-subscribers) are inherently excluded from the federation, since the goal of the server is to optimize the model for the priority clients only. In this context, FedALIGN can be seen as a way to attain higher fairness in the PFL setting. To explain this note if one were to utilize FedAvg in PFL, there would be two possible options: (i) using FedAvg only on the priority clients and none of the non-priority clients (say FedAvg-None) or (ii) using FedAvg on all priority and non-priority clients (say FedAvg-All).

A natural definition of fairness in this case is the fraction of the overall population of clients that has access to a model that is useful to them. This definition of fairness used is inspired by many works Zafar et al. (2017); Mohri et al. (2019); Li et al. (2020b), but is adapted to be more relevant to the PFL setting. Under this definition, (i) would lead to little fairness (but better model performance for priority clients), and (ii) would lead to higher fairness, but worse model performance, again for priority clients. In real-world settings (ii) will not be accepted by priority clients, since they do not stand to benefit by including non-priority clients. Therefore FedALIGN attains higher fairness in the PFL setting (although not as high as a full FedAvg-All, which is not realistic). The priority clients stand to benefit from this (as they will ultimately receive a better model) and will agree to the model being shared with non-priority clients, and non-priority clients stand to benefit (at least some fraction of them) from gaining access to a well-trained model, even though they are not subscribers/priority-clients. Thus FedALIGN naturally leads to the most realistically fair algorithm for the subscription based setting we model.

## 2 Problem Setting and Related Works

In PFL, we assume that there are $N$ clients, and client $k$ has the local dataset $\mathcal{B}_k$, with $|\mathcal{B}_k| = D_k$. The clients are split into two sets: priority clients $\mathcal{P} \subset [1:N]$ and non-priority clients. All clients are connected to a central server, who wishes to minimize the objective

$$\sum_{k \in \mathcal{P}} p_k F_k(\mathbf{w}), \tag{5}$$

where $p_k := D_k / \sum_{i \in \mathcal{P}} D_i$ is the fraction of data at the $k$th client and $F_k(\mathbf{w}) := \frac{1}{D_k} \sum_{\xi \in \mathcal{B}_k} f(\mathbf{w}; \xi)$ is the local objective function of the $k$th client. Note that since we normalize the data with the total data possessed only by the priority clients, $\sum_{k \in \mathcal{P}} p_k = 1$, but $\sum_{k=1}^{N} p_k \neq 1$ in general. We denote $t$ to indicate the number of local updates. In each communication round, each client that participates in that round, independently runs $E$ local epochs. Aggregation happens when $t \pmod{E} = 0$.

**Other Settings with Client Prioritization.** The need to prioritize a subset of clients may arise in contexts other than subscription services discussed in Section 1. For instance, existing FL settings that focus on optimizing objectives defined by their clients may face challenges when integrating additional clients. While these new clients may be valuable to the FL setting, their inclusion often necessitates a shift in the learning objective. FedALIGN provides a partial solution to this predicament, enabling the inclusion of well-aligned clients without necessitating alterations to the learning objective.

Another context that naturally leads to prioritization of some clients, is when some clients are slow, unreliable, or resource-poor. This leads to the "straggler effect", which can significantly hamper the overall learning process (Wang et al., 2020a; Hard et al., 2019; Nishio and Yonetani, 2019). The straggler effect occurs because the global model update must wait until slower clients complete their local computations, and the overall update time is when the overall computation time is dictated by the slowest participants. In many scenarios it is usually known which clients are likely to be slow. An example is households with multiple smart devices, where models are trained with a few powerful devices like smart phones and laptops, with additional less powerful devices like IOT devices (with potentially different data) also available for computation. In this case it may be fruitful to exclude stragglers from the global objective and only include their updates if they are well-aligned to this objective. Moreover in our supplementary material we extend our convergence analysis to the case where non-priority clients are free to participate in aggregation as they want to, naturally modelling stragglers who may only be able to provide updates in a few rounds.

**Selfish Federated Learning.** To the best of our knowledge, only one work (Anonymous, 2022) has explored a model similar to ours, which they call Selfish Federated Learning. In the Selfish paradigm, it is assumed that non-priority clients always contribute to the federation irrespective of what global model they receive, an assumption that may not hold true in practice, since clients need to be incentivized to participate. Secondly, the previous model assumes that the number of priority clients are much smaller than the total clients. This assumption can be overly restrictive, limiting the model's applicability in varied scenarios. Furthermore, their algorithm exhibits convergence only under highly stringent learning rates, thereby further confining its practical utility. Furthermore, regarding client participation, the model requires involvement from all non-priority clients in each communication round. This expectation is impractical and significantly impacts communication efficiency. Lastly, it presumes identical data distribution among priority clients, which essentially aligns it with the framework of Personalized FL, rendering the premise of their approach somewhat trivial. Our work, in contrast, imposes no such assumptions, but instead looks at a more pragmatic scenario as outlined in the introduction. While an other study (Chayti et al., 2021) has investigated scenarios where clients not involved in the global objective can still contribute to its improvement, these works necessitate full participation. These approaches are incompatible with federated learning contexts. Our model and algorithm naturally addresses these challenges.

**Personalized FL.** Various FL strategies have been explored to create personalized models. One common approach involves adapting a global FL model to each local client's specific requirements (Wang et al., 2020b; Yang et al., 2021; Chai et al., 2020; Smith et al., 2018), while another leverages inter-client relationships to fine-tune personalized models, often achieved by introducing adjustable regularization terms into each client's local objective (Kairouz et al., 2019; Karimireddy et al., 2021; Cheng et al., 2021). In these scenarios, a universally applicable model is usually maintained, which is subsequently refined to suit individual clients'

local data. Furthermore, clustering techniques (Sattler et al., 2019) have been employed, wherein clients are categorized based on the similarity of their data, leading to the development of models tailored to each cluster's unique characteristics. The end goal remains the same: the customization of models for each participating client during the aggregation phase. For a comprehensive review of personalized FL, we refer to the survey in Tan et al. (2022). Our approach deviates from these strategies in two critical ways: (1) Our global objective is constructed solely from a pre-selected subset of clients, who might exhibit substantial heterogeneity, and (2) Our training does not explicitly weigh the importance of non-priority clients. Instead, we train a single model that inherently benefits non-priority clients that are well-aligned.

**Tackling Heterogeneity in FL.** Many methods aim to tackle data heterogeneity in FL. FedProx (Kairouz et al., 2019) adds proximal terms to local objectives and SCAFFOLD (Karimireddy et al., 2021) tackles client drift. There are many other relevant works which aim to tackle heterogeneity (Reddi et al., 2020; Haddadpour and Mahdavi, 2019; Khaled et al., 2019; Stich and Karimireddy, 2021; Woodworth et al., 2020; Koloskova et al., 2021; Zhang et al., 2021; Pathak and Wainwright, 2020). These methods aims to tune a global model to work well on all local clients. FedALIGN also targets heterogeneity, but specifically tries to reduce the heterogeneity for priority clients by leveraging non-priority clients. The selection approach of FedALIGN may be combined with other algorithms to improve their performance in this setting.

**Client Selection in Federated Learning.** Several existing methods employ sample re-weighting strategies based on individual client utility. Some approaches define client utility based on the norm of gradients (Zeng et al., 2021), while others leverage local loss values as the utility measure (Chen et al., 2020). One particular study biases weights in favor of clients with higher losses for faster convergence towards the global objective, albeit at the expense of an enduring bias term (Cho et al., 2020). Alternatively, some approaches increase weights on clients with lower losses, mitigating the risk of clients with noisy data adversely influencing the global model (Ghadikolaei et al., 2019). Our approach differs from these strategies. We employ a loss matching strategy, wherein non-priority clients only share models when the received global model yields sufficiently low loss on their local dataset.

While these client selection algorithms may provide ideas for algorithm development in PFL, they are designed for the classical setting of FL and do not provide theoretical guarantees for PFL and are likely ill-suited for PFL.

Table 1: Comparison of Client Selection Strategies

| Method | Server level decisions | Loss-based Utility | Gradient-based Utility |
|---|---|---|---|
| Zeng et al. (2021) | ✓ | ✗ | ✓ |
| Chen et al. (2020) | ✓ | ✓ | ✗ |
| Cho et al. (2020) | ✓ | ✓ | ✗ |
| Ghadikolaei et al. (2019) | ✓ | ✗ | ✓ |
| Wang et al. (2020a) | ✓ | ✓ | ✗ |
| Fu et al. (2023) | ✓ | ✗ | ✓ |
| FedALIGN (Our Method) | ✗ | ✓ | ✗ |

While client selection as a concept is similar to our problem setting, the methods used for client selection in classical settings often use updates from all the clients while making decisions, which is unrealistic in the PFL setting. In FL, there are two kinds of communication: downstream from server to clients and upstream from clients to server Sattler et al. (2020). In FedALIGN, the server broadcasts downstream to all clients, while only aligned non-priority clients communicate upstream. This is different from current methods, which typically require full communication from all clients. This makes FedALIGN communication effcient in the PFL setting. For example, Wang et al. (2020a) presents a loss-based selection algorithm by picking clients with the highest loss. This however requires you to make server level decisions on which client has the highest loss, implying that the non-priority clients would all be participating in every communication round, which is unrealistic in the PFL setting, and FedALIGN is specifically designed to not require all non-non-priority clients to participate. Similarly, there are other works involving gradient-based selection criteria (Fu et al., 2023), which also require such server-level decisions.

Notably, gradient comparisons are more effective than loss comparisons since gradient alignment is a better indicator of model alignment at a given time (Zeng et al., 2021). Moreover, absolute loss based methods like those proposed by Cho et al. (2020) offer an advantage in terms of efficiency, as they eliminate the bottleneck caused by repeatedly determining the client with the lowest loss. Sorting clients based on absolute loss values, rather than comparison with relative loss values like FedALIGN, is computationally more efficient. It is thus important to acknowledge that other methods have their own advantages and may be more useful in different contexts. For example, absolute loss-based methods Cho et al. (2020); Chen et al. (2020); Ghadikolaei et al. (2019) can be particularly effective in scenarios where rapid convergence is critical, such as in time-sensitive applications where communication is not the bottleneck, but computation is. Thus, while our approach offers significant benefits in terms of communication efficiency (since we do not require server level decisions) in the PFL setting, specific methods may prove more advantageous in other scenarios such computational efficiency.

## 3 FedALIGN and Convergence

In this section we describe FedALIGN and prove its convergence guarantee. Our results are presented for the full client participation setting, though the analysis can be easily extended to the case where a subset of clients (both priority and non-priority) are picked uniformly at random. Further we can extend the result for a more general case, where non-priority clients follow an arbitrary participation rule or are given the choice to participate in any given round. This is relevant in the stragglers setting, for example. These cases are considered in the supplementary material.

### 3.1 FedALIGN

FedALIGN introduces a simple decision rule to select clients based on the relative discrepancy between local and global loss. Non-priority clients are conditionally included in the aggregation process during a communication round corresponding to the $t$th local round, if for the provided global model $\mathbf{w}_t$, and a chosen threshold $\epsilon_t$ , the absolute value of the difference between a non-priority client's local loss and the global loss is below $\epsilon_t$; i.e., $|F(\mathbf{w}_t) - F_k(\mathbf{w}_t)| \leq \epsilon_t$. This method ensures that updates contributing to the global model do not diverge significantly from the overarching learning objective, which is determined by the priority clients. By strategically aligning the updates from non-priority clients with the global learning objective, FedALIGN can effectively harness the collective learning power of the entire network, while maintaining focus on the primary learning task. The exact steps taken are provided in Algorithm 1.

In practice, during a communication round, along with the model parameter $\mathbf{w}_t$, the server sends the value of accuracy on the global data. The non-priority clients send back updates, only if the accuracy (accuracy is used as a proxy for loss) of the received model on it's local data is high enough. This is done to ensure that clients only communicate when they are well-aligned. The server then decides to aggregate it to the global model if it is well aligned, based on how close the accuracies are; i.e., if the full condition $|F(\mathbf{w}_t) - F_k(\mathbf{w}_t)| \leq \epsilon_t$ is satisfied.

### 3.2 Convergence Analysis

Before stating the main theoretical result on the convergence of FedALIGN, we present key assumptions and notation. We make the following assumptions on local objective functions $F_1, \ldots, F_n$. Assumptions 1 and 2 are standard in optimization, while Assumptions 3 and 4 are standard in distributed learning and FL literature (Li et al., 2020a; Cho et al., 2020; Zhang et al., 2013; Stich, 2019).

**Assumption 1.** *(Smoothness) For each $f \in \{F_1, \ldots, F_N\}$, $f$ is $L$-smooth, that is, for all $\mathbf{u}$ and $\mathbf{v}$, $f(\mathbf{v}) \leq f(\mathbf{u}) + (\mathbf{v} - \mathbf{u})^T \nabla f(\mathbf{u}) + \frac{L}{2}\|\mathbf{v} - \mathbf{u}\|^2$.*

**Assumption 2.** *(Convexity) For each $f \in \{F_1, \ldots, F_N\}$, $f$ is $\mu$-strongly convex, that is, for all $\mathbf{u}$ and $\mathbf{v}$, $f(\mathbf{v}) \geq f(\mathbf{u}) + (\mathbf{v} - \mathbf{u})^T \nabla f(\mathbf{u}) + \frac{\mu}{2}\|\mathbf{v} - \mathbf{u}\|^2$.*

**Assumption 3.** *(Unbiased and Bounded Variance of Gradients) For a mini-batch $\xi_k$ randomly sampled from $B_k$ from user $k$, the SGD is unbiased; i.e., $\mathbb{E}[\nabla F_k(\mathbf{w}_k, \xi_k)] = \nabla F_k(w_k)$. Also, the variance of random gradients is bounded: $\mathbb{E}[\|\nabla F_k(\mathbf{w}_k, \xi_k) - \nabla F_k(\mathbf{w}_k)\|^2] \leq \sigma^2$ for $k = 1, \ldots, N$.*

---

**Algorithm 1:** FedALIGN: Adaptive Client Selection in Prioritized Federated Learning

---

**Input:** Data Fraction: $\mathcal{P}$, Communication Rounds: $T$, Initial model: $\mathbf{w}_0$, Threshold: $\epsilon_t \forall t \in [1:T]$,
       Local Epochs: $E$

**Output:** $\mathbf{w}_T$

$\mathbf{w}_t \leftarrow \mathbf{w}_0$

**for** each communication round $t = 0, E, \dots$ **do**

    `// Server executes:`

    $p \leftarrow 1$ // data fraction

    Broadcast $\mathbf{w}_t, F(\mathbf{w}_t)$ to all clients

    $\mathbf{w}_{t+E} \leftarrow 0$

    **for** *each client $k \in \mathcal{P}$* **do**

        `// Client k executes:`

        $\mathbf{w}_{t+E}^k \leftarrow \text{ClientUpdate}(k, \mathbf{w}_t)$

        `// Server executes:`

        $\mathbf{w}_{t+E} \leftarrow \mathbf{w}_{t+E} + p_k \mathbf{w}_{t+E}^k$

    **end**

    **for** each client $k \notin \mathcal{P}$ **do**

        `// Client k executes:`

        Compute local loss $F_k(\mathbf{w}_t)$

        **if** $F_k(\mathbf{w}_t) \leq F(\mathbf{w}_t) + \epsilon_t$ **then**

            $\mathbf{w}_{t+E}^k \leftarrow \text{ClientUpdate}(k, \mathbf{w}_t)$

            **return** $F_k(\mathbf{w}_t)$

            `// Server executes:`

            **if** $F_k(\mathbf{w}_t) \geq F(\mathbf{w}_t) - \epsilon_t$ **then**

                $\mathbf{w}_{t+E} \leftarrow \mathbf{w}_{t+E} + p_k \mathbf{w}_{t+E}^k$

                $p \leftarrow p + p_k$

            **end**

        **else**

            Remain silent for this round

        **end**

    **end**

    `// Server executes:`

    $\mathbf{w}_{t+E} \leftarrow \mathbf{w}_{t+E}/p$

**end**

**return** $\mathbf{w}_T$

**Function** `ClientUpdate`($k$, $\mathbf{w}$, $\eta$, $E$, $b$)

    $\mathcal{B}_k \leftarrow$ local dataset of client $k$

    $B \leftarrow$ (split $\mathcal{B}_k$ into batches of size $b$)

    **for** $i = 1, \dots, E$ **do**

        **for** *each batch $j \in B$* **do**

            $\mathbf{w} \leftarrow \mathbf{w} - \eta \nabla F_k(\mathbf{w})$

        **end**

    **end**

    **return** $\mathbf{w}$

---

Note that in general we can assume that each client has a distinct variance bound $\sigma_k$.

**Assumption 4.** *(Non-Divergent Gradients) The stochastic gradient's expected squared norm is uniformly bounded, i.e., $\mathbb{E}[\|\nabla F_k(\mathbf{w}_k, \xi_k)\|^2] \leq G^2$ for $k = 1, \dots, N$.*

As in (2), we define $\Gamma := F^* - \sum_{k \in \mathcal{P}} p_k F_k^*$ to be the heterogeneity parameter, but here only defined for the priority clients $\mathcal{P}$. A larger $\Gamma$ implies a higher degree of heterogeneity. When $\Gamma = 0$, the local objectives and global objectives are consistent. Recall that the index $t$ counts the number of local updates, and each

client runs $E$ local updates before sending the updates when $t \pmod{E} = 0$. Moreover we also define $\tau(t) := \max\{t' : t' \pmod{E} = 0, t' \leq t\}$ as the time at which the last communication round occurred. Further define $\Gamma_k := F_k(\mathbf{w}^*) - F_k^*$, which is the objective gap between the optimal global model and the optimal local model.

Here we present the convergence result for FedALIGN in the full participation scenario (partial participation is considered in the appendix) and compare it to existing results. Suppose FedALIGN runs for $T$ iterations, for a $T$ such that $T \pmod{E} = 0$ and outputs $\mathbf{w}_T$ as the global model.

**Theorem 1.** *(Convergence) Suppose that Assumptions 1, 2, 3, and 4 hold, and we set a decaying learning rate $\eta_t = \frac{2}{\mu(t+\gamma)}$, where $\gamma = \max\left\{\frac{8L}{\mu}, E\right\}$ and any $\epsilon_t \geq 0$. The expected error following $T$ total local iterations (equivalent to $T/E$ communication rounds) for* FedALIGN *under full device participation satisfies*

$$\mathbb{E}[F(\mathbf{w}_T)] - F^* \leq \frac{1}{T+\gamma}\left(C_1 + C_2\theta_T\Gamma\right) + \rho_T, \tag{6}$$

*where $\theta_T \in [0,1]$ and $\rho_T$ can be adjusted by selecting an appropriate $\epsilon_t$ in each time step ($\rho_T$ and $\theta_T$ are defined in paragraphs after the theorem statement). Here, $C_1$ and $C_2$ are constants given by*

$$C_1 := \frac{2L}{\mu^2}\left(\sigma^2 + 8(E-1)^2G^2\right) + \frac{4L^2}{\mu}\|\mathbf{w}_0 - \mathbf{w}^*\|^2,$$

$$C_2 := \frac{12L^2}{\mu^2}.$$

This theorem is proved in the supplementary material. We now define and explain some of the important constants we introduce in the result.

**Alignment of non-priority clients $\theta_T$.** The expression for $\theta_T$ is

$$\theta_T := \mathbb{E}\left[\frac{1}{T+\gamma-2}\sum_{i=1}^{T-1}\frac{1}{1+\sum_{k\notin\mathcal{P}}p_k I_{k,\tau(i)}}\right], \tag{7}$$

where $I_{k,t} := \mathbf{1}\{|F_k\left(\mathbf{w}_t^k\right) - F\left(\mathbf{w}_t^k\right)| < \epsilon_t\}$ is an indicator random variable for whether a non-priority client is included in a given communication round. The alignment of non-priority clients, $\theta_T$, effectively measures the average inclusion of non-priority clients over $T-1$ time steps. Lower values of $\theta_T$ correspond to a greater number of clients being included in aggregation. Specifically the convergence rate improves to $O(\theta_T/T)$ in each time step. This kind of improvement due to client selection strategies has been considered before (Cho et al., 2020).

**Tunable bias $\rho_T$.** The expression for $\rho_T$ is

$$\rho_T := \frac{2L}{\mu(T+\gamma-2)}\sum_{i=1}^{T-1}\mathbb{E}\left[\frac{\sum_{k\notin P}p_k I_{k,\tau(i)}\Gamma_k}{1+\sum_{k\notin\mathcal{P}}p_k I_{k,\tau(i)}}\right]. \tag{8}$$

This term is an average aggregation of the bias introduced in the first $T-1$ time steps.

The parameters $\theta_T$ and $\rho_T$ effect the convergence of the learning model under consideration, in opposite ways. The choice of $\epsilon_t$ at each time step plays a crucial role in controlling this balance. If $\epsilon_t$ is set too low, it risks under-utilizing the potential of well-aligned clients, thereby causing a slower rate of convergence. On the other hand, if $\epsilon_t$ is set too high, it introduces a high bias in the aggregated model updates, which can negatively impact the model's performance.

**Optimality of FedALIGN.** The above dichotomy highlights the fine-tune (Cheng et al., 2021) aspect of the FedALIGN algorithm. We can thus set $\epsilon_t$ such that it is gradually reduced in the later rounds. This strategy ensures that the algorithm initially benefits from the contributions of a larger number of clients (albeit potentially introducing some bias), but as the rounds progress and the model is closer to convergence, the bias can gradually be eliminated by reducing $\epsilon_t$. This enables the algorithm to leverage the power of well-aligned

clients to expedite convergence initially, and then gradually reduces the bias to ensure accurate model learning. Specifically we can ensure that FedALIGN will converge to a global optimum as $\epsilon_T$ approaches 0 when $T$ tends to infinity.

**Corollary 2.** *(Optimal Convergence) Assuming that the conditions in Theorem 1 hold and that $\epsilon_T \to 0$ as $T \to \infty$, we have*

$$\lim_{T \to \infty} E[F(\mathbf{w}_T)] = F^*. \tag{9}$$

Corollary 2 is a straightforward consequence of Theorem 1 and is proved in the supplementary material. Further, in the supplementary material, we conduct experiments that set $\epsilon_T$ to be distinct monotonically decreasing functions of $T$. We use our benchmark datasets to evaluate the effect of different decay rates of $\epsilon_T$ on convergence.

**Consistency of our result with FedAvg on priority clients**. If we set $\epsilon_t = 0$ for all time steps in FedALIGN, $\theta_T$ (the alignment of non-priority clients) becomes 1, implying that all priority clients are included in the aggregation. At the same time, $\rho_T$ (the tunable bias) becomes 0, indicating that no bias is introduced in the model updates. Under these conditions, FedALIGN is identical to FedAvg on just the priority clients, and the convergence bound is precisely the result in Li et al. (2020a). It shows how FedALIGN builds upon FedAvg by introducing additional flexibility and control through the $\epsilon_t$ parameter, which allows us to manage the trade-off between bias and speed of convergence. We also highlight that we can modify other algorithms like FedPROX (Kairouz et al., 2019) to obtain a similar control. We show experimental results in our supplementary material to validate this.

## 4 Experiments

We conduct experiments on various benchmark datasets: FMNIST, *balanced* EMNIST and CIFAR-10 (Xiao et al., 2017). The datasets were preprocessed following the methods developed in Brendan McMahan et al. (2017). For FMNIST we use a logistic regression model, for EMNIST we use a two-layer neural network and for the CIFAR-10 dataset we use a CNN. The dataset is generated by distributing uni-class shards to each client, following the steps in Brendan McMahan et al. (2017). More details about the architectures are deferred to the supplementary material. In addition, in order to assess the effect of noisy and irrelevant data in the non-priority clients, we modify the SYNTH$(1, 1)$ dataset considered in Li et al. (2018). We use a logistic regression model for this experiment.

We evaluate FedALIGN (built on FedAvg) against two baselines: (1) FedAvg on only the priority clients and (2) FedAvg on all clients. Further in our supplementary material we consider FedALIGN built on FedPROX (Li et al., 2018). We consider the full participation case, where all clients are sent a global model by the server and consider the case of partial participation (where only a randomly sampled subset of clients are chosen in each round) in the supplementary material.

**Experiment results on benchmark data**. For the learning rate ($\eta$), we adopted a value of 0.1 for both FMNIST and EMNIST datasets, whereas for CIFAR-10, we utilized a lower learning rate of 0.01. The selection of these learning rates was made after conducting a grid search of different learning rates on small sets of the data. The selected learning rates resulted in stable convergence and exhibited the best performance. All experiments were conducted with 5 different seeds for randomness, setting the local epoch $E = 5$. We expand on the impact of varying epoch choices in the supplementary material. The initial 10% of communication rounds were dedicated as warm-up rounds during which only priority clients contributed to the aggregation process. This design was to ensure the development of a reasonably working model prior to the inclusion of non-priority clients. We found by optimizing over a grid of different $\epsilon$'s that the selection $\epsilon = 0.2$ and eventually reducing it to 0, performs the best. The results depicted in Figure 1 show the superiority of FedALIGN over the baseline methods. This is evident in the post-warm-up phase, where there is a noticeable acceleration in convergence rates. Interestingly, it was observed that the inclusion of all clients often led to a degradation in overall performance, further affirming the importance of selecting well-aligned clients. We further explore alternative configurations of priority client selections with varying data fractions in the supplementary material.

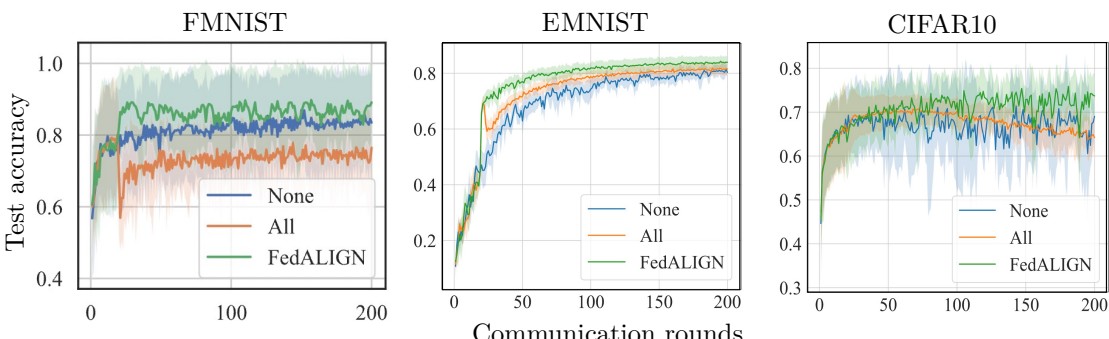

Figure 1: Test accuracy for benchmark datasets, under full participation, with 2 heterogeneous priority clients and $N = 60$ clients, with $E = 5$. After the first 20 warm-up rounds, a clear increase in convergence rate is observed for FedALIGN (green), while a deterioration is observed for All (orange) (FedAvg with all clients). FedALIGN also achieves a higher final accuracy.

**Experimental results on synthetic data.** We simulate scenarios to analyze the performance of non-priority clients with varying degrees of alignment. For this purpose, we utilize the Synth$(1, 1)$ dataset as referred to in Li et al. (2018). We create a global dataset and distibute this data to non-priority clients and add two forms of noise: (1) Discrepancies in label assignments for data points, and (2) the addition of irrelevant data points, which are generated by independent distributions. We configure the noise to ensure that the non-priority clients experience varying degrees of noise, thereby influencing their alignment with the objective. The detailed methodology for the generation of this noise is provided in the supplementary materials. We apply a logistic regression model to this data, maintaining a learning rate of 0.1, with local models used to find the best and stable learning rates, as in the prior experiments. We consider 3 different average noise/discrepancy levels across non-priority clients: medium, low and high. We found that $\epsilon = 0.2$ is the best choice for the selection parameter for low and medium noise, while $\epsilon = 0.4$ is the best choice for the high noise experiment. From Figure 2, it becomes evident that FedALIGN consistently outperforms baselines, under this more complicated form of misalignment in the non-priotity clients. This underscores the robustness of FedALIGN even in challenging conditions.

## 5   Concluding Remarks

We proposed Prioritized Federated Learning, a FL setting that models the real-world situation where some clients are given priority from the point of view of model training, but other clients may choose to participate in the federation as well. We introduced a simple algorithm (FedALIGN), that smartly selects non-priority clients to help achieve the goals of priority clients. This creates a win-win situation where non-priority clients that align well with the system obtain a strong global model that can enhance their local tasks. The convergence analysis of FedALIGN highlights an important trade-off: using more priority clients can speed up convergence but introduces a bias. We show that this bias can be eliminated in later rounds. Our extensive experiments support these findings.

**G**eneralized PFL. One can conceptualize traditional FL and PFL as two ends of a spectrum. On one end, traditional FL involves weighing the objectives generated by both priority and non-priority clients according to their respective data fractions. This ensures that each client, regardless of priority status, contributes proportionally to the overall model training. On the other end of the spectrum, PFL focuses exclusively on the objectives generated by priority clients, effectively assigning a weight of zero to the non-priority clients. This approach prioritizes the specific needs and data characteristics of the priority clients, ensuring that the model is tailored to their unique requirements. An intermediate approach would involve assigning varying weights to the objectives generated by priority and non-priority clients, with a greater emphasis on the priority clients. By adjusting these weights, we can control the influence of each client group on the model training process. This allows for a flexible and nuanced approach, balancing the benefits of penalization for

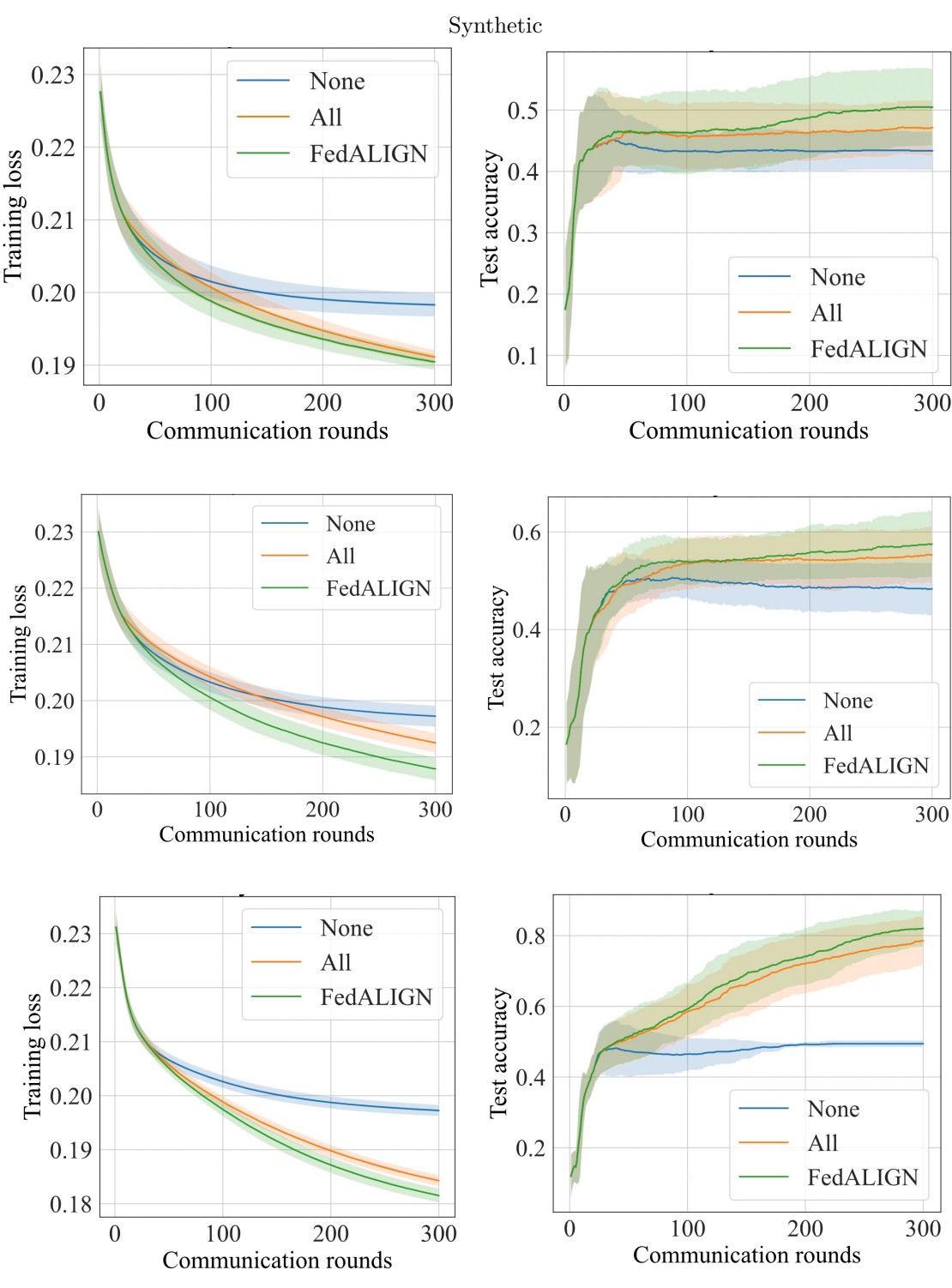

Figure 2: Training loss and test accuracy for the SYNTH(1,1) dataset, under full participation, with 10 heterogeneous priority clients and $N = 20$ clients, with $E = 5$. The first graph considers the case where there is medium noise, the second considers the low noise case and the third considers the high noise case. After the first 20 warm-up rounds, a clear improvement is observed for FedALIGN (green) for all these cases.

priority clients with the broader generalization capabilities of traditional FL. Exploring this spectrum and adjusting the weights accordingly presents an exciting direction for future research.

**On the willingness of non-priority clients to participate.** We argue that the receipt of a well-performing global model is one of many incentives for non-priority clients to contribute to the aggregation. In our supplementary material, we experimentally show numerous instances where the globally trained model outperforms locally trained counterparts. During instances where the received model does not surpass the performance of a locally trained model, primarily due to the latter's fine-tuned adaptation to specific client characteristics, we posit that globally trained models help refine local models. This concept of leveraging external data for local fine-tuning has been the subject of extensive research in Personalized FL (Tan et al., 2022). Furthermore, we assert that access to a global model that demonstrates satisfactory performance on local client data is invariably beneficial. It contributes to the suite of tools available to the local client to potentially improve overall performance. A compelling future direction is thus to formalize these incentives as utility functions.

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
