## A  Proof of Theorem 1

**Theorem A.1.** *(Convergence) Suppose that Assumptions 1, 2, 3, and 4 hold, and we set a decaying learning rate $\eta_t = \frac{2}{\mu(t+\gamma)}$, where $\gamma = \max\left\{\frac{8L}{\mu}, E\right\}$ and any $\epsilon_t \geq 0$. The expected error following $T$ total local iterations (equivalent to $T/E$ communication rounds) for* FedALIGN *under full device participation satisfies*

$$\mathbb{E}[F(\mathbf{w}_T)] - F^* \leq \frac{1}{T+\gamma}\left(C_1 + C_2\theta_T\Gamma\right) + \rho_T, \tag{10}$$

*where $\theta_T \in [0,1]$ and $\rho_T$ can be adjusted by selecting an appropriate $\epsilon_t$ in each time step ($\rho_T$ and $\theta_T$ are defined after the theorem statement). Here, $C_1$ and $C_2$ are constants given by*

$$C_1 := \frac{2L}{\mu^2}\left(\sigma^2 + 8(E-1)^2G^2\right) + \frac{4L^2}{\mu}\|\mathbf{w}_0 - \mathbf{w}^*\|^2, \quad C_2 := \frac{12L^2}{\mu^2}.$$

Here the terms $\theta_T$ and $\rho_T$ are defined as

$$\theta_T = \frac{1}{T+\gamma-2}\sum_{i=1}^{T-1}\mathbb{E}\left[\frac{1}{1+\sum_{k\notin\mathcal{P}}p_kI_{k,\tau(i)}}\right], \tag{11}$$

and

$$\rho_T := \frac{2L}{\mu(T+\gamma-2)}\sum_{i=1}^{T-1}\mathbb{E}\left[\frac{\sum_{k\notin P}p_kI_{k,\tau(i)}\Gamma_k}{1+\sum_{k\notin\mathcal{P}}p_kI_{k,\tau(i)}}\right]. \tag{12}$$

We analyze FedALIGN in the full device participation setting here. The proof skeleton here follows Cho et al. (2020), but considering a selection criteria for non-priority clients in each communication round requires novel analysis.

### A.1  Some definitions and preliminaries

First we define the *re-normalized* data fraction as

$$p_k'(t) := \frac{p_k}{1+\sum_{k\notin\mathcal{P}}p_kI_{k,\tau(t)}}. \tag{13}$$

Generally, $p_k'(t)$ is a random variable, dependent on $(\epsilon_{\tau(t)}, \tau(t))$, and can assume a distinct value in each communication round. In the proof, we sometimes express $p_k'(t)$ as $p_k'$, which is a minor notational liberty.

Note that our aggregation rule is as follows: During every communication round, the global model $\mathbf{w}_{\tau(t)}$ aggregates the weighted sum of the local models, renormalized by total data contributed by all clients included in that communication round. This update is given by

$$\mathbf{w}_{\tau(t)} \leftarrow \frac{1}{1+\sum_{k\notin\mathcal{P}}p_kI_{k,\tau(t)}}\left(\sum_{k\in\mathcal{P}}p_k\mathbf{w}_{\tau(t)}^k + \sum_{k\notin\mathcal{P}}p_k\mathbf{w}_{\tau(t)}^kI_{k,\tau(t)}\right).$$

This aggregated expression rewritten in terms of $p_k'$ has the following form:

$$\sum_{k\in\mathcal{P}}p_k'\mathbf{w}_{\tau(t)}^k + \sum_{k\notin\mathcal{P}}p_k'\mathbf{w}_{\tau(t)}^kI_{k,\tau(t)}. \tag{14}$$

This motivates us to define a "virtual" sequence

$$\mathbf{w}_t = \sum_{k\in\mathcal{P}}p_k'\mathbf{w}_t^k + \sum_{k\notin\mathcal{P}}p_k'\mathbf{w}_t^kI_{k,\tau(t)}, \tag{15}$$

which intuitively captures an aggregation (if it happens) in every local round. The value of $\mathbf{w}_t$ is indeed the aggregated global model when $t$ corresponds to a communication round. Now, define the aggregated stochastic gradients as

$$\mathbf{g}_t = \sum_{k=1}^{N} p'_k \nabla F_k(\mathbf{w}_t^k, \xi_t) I_{k,\tau(t)}, \tag{16}$$

and the aggregated gradients as

$$\overline{\mathbf{g}}_t = \sum_{k=1}^{N} p'_k \nabla F_k(\mathbf{w}_t^k) I_{k,\tau(t)}, \tag{17}$$

where above, for ease of analysis, we redefine $I_{k,\tau(t)}$ to be

$$I_{k,\tau(t)} := \mathbf{1}\{|F_k\left(\mathbf{w}_t^k\right) - F\left(\mathbf{w}_t^k\right)| < \epsilon_t, \text{ or } k \text{ is the index of a priority client}\}. \tag{18}$$

This definition also allows us to rewrite $\mathbf{w}_t$ as

$$\mathbf{w}_t := \sum_{k=1}^{N} p'_k \mathbf{w}_t^k I_{k,\tau(t)}. \tag{19}$$

Both definitions are equivalent, and we will use either definition as required. We also use $\mathbb{E}$ to mean expectation over all sources of randomness for a fixed time $t$, while $\mathbb{E}_{\cdot|p'_k, I_{k,\tau(t)} \forall k}$ refers to the conditional expectation, conditioned on random variables $(p'_k, I_{k,\tau(t)} \forall k)$.

We now state some preliminary lemmas. We state more directly relevant lemmas in the next section.

**Lemma A.2.** *(Gradient is $L$-close to its minima) If $F_k$ is $L$-smooth with global minimum value $F_k^*$, then for any $\mathbf{w}_k$, we have*

$$\|\nabla F_k(\mathbf{w}_k)\|^2 \le 2L(F_k(\mathbf{w}_k) - F_k^*). \tag{20}$$

*Proof.* By definition of $L$-smoothness, we have

$$F_k^* \le F_k(\mathbf{v}) \le F_k(\mathbf{w}_k) + (\mathbf{v} - \mathbf{w}_k)^T \nabla F_k(\mathbf{w}_k) + \frac{L}{2}\|\mathbf{v} - \mathbf{w}_k\|^2.$$

Since this is true for any $\mathbf{v}$, one can set $\mathbf{v} = \mathbf{w}_k - (1/L)\nabla F_k(\mathbf{w}_k)$. This gives us

$$F_k^* \le F(\mathbf{w}_k) - \frac{1}{L}[\nabla F_k(\mathbf{w}_k)]^T \nabla F_k(\mathbf{w}_k) + \frac{1}{2L}\|\nabla F_k(\mathbf{w}_k)\|^2 = F_k(\mathbf{w}_k) - \frac{1}{2L}\|\nabla F_k(\mathbf{w}_k)\|^2,$$

or

$$\|\nabla F_k(\mathbf{w}_k)\|^2 \le 2L(F_k(\mathbf{w}_k) - F_k^*). \tag{21}$$

$\square$

**Lemma A.3.** *(Aggregated model is closer to $\mathbf{w}^*$ than local models) Given local models $\mathbf{w}_t^k$ and the global minima $\mathbf{w}^*$ of $F(\cdot)$,*

$$\mathbb{E}\left[\|\mathbf{w}_t - \mathbf{w}^*\|^2\right] \le \mathbb{E} \sum_{k=1}^{N} p'_k I_{k,\tau(t)} \mathbb{E}_{\cdot|p'_k I_{k,\tau(t)}} \|\mathbf{w}_t^k - \mathbf{w}^*\|^2. \tag{22}$$

*Proof.* The lemma follows because

$$\mathbb{E}\left[\|\mathbf{w}_t - \mathbf{w}^*\|^2\right] = \mathbb{E}\left[\left\|\sum_{k=1}^{N} p'_k I_{k,\tau(t)} \mathbf{w}_k^t - \mathbf{w}^*\right\|^2\right] \overset{(a)}{=} \mathbb{E}\left[\left\|\sum_{k=1}^{N} p'_k I_{k,\tau(t)}(\mathbf{w}_k^t - \mathbf{w}^*)\right\|^2\right]$$

$$\le \mathbb{E} \sum_{k=1}^{n} p'_k I_{k,\tau(t)} I_{k,\tau(t)} \|\mathbf{w}_t^k - \mathbf{w}^*\|^2 \le \mathbb{E} \sum_{k=1}^{n} p'_k I_{k,\tau(t)} \mathbb{E}_{\cdot|p'_k I_{k,\tau(t)}} \|\mathbf{w}_t^k - \mathbf{w}^*\|^2, \tag{23}$$

where $(a)$ is because $\sum_{k=1}^{N} p'_k I_{k,\tau(t)} = 1$, from the definitions of $p'_k$ and $I_{k,\tau(t)}$. $\square$

**Lemma A.4.** *(Average discrepancy caused by local updates.) Given Assumption 4, for $\eta_{\tau(t)} \leq 2\eta_t$ and all $t - \tau(t) \leq E - 1$, we have*

$$\mathbb{E} \sum_{k=1}^{N} p'_k \|\mathbf{w}_t^k - \mathbf{w}_t\|^2 I_{k,\tau(t)} \leq 4\eta_t^2 (E-1)^2 G^2. \tag{24}$$

*Proof.* We have that

$$
\begin{aligned}
\mathbb{E} \sum_{k=1}^{N} p'_k \|\mathbf{w}_t^k - \mathbf{w}_t\|^2 I_{k,\tau(t)} &= \mathbb{E} \sum_{k=1}^{N} p'_k \left\| \left(\mathbf{w}_t^k - \mathbf{w}_{\tau(t)}\right) - \left(\mathbf{w}_t - \mathbf{w}_{\tau(t)}\right) \right\|^2 I_{k,\tau(t)} \\
&\stackrel{(a)}{\leq} \mathbb{E} \sum_{k=1}^{N} p'_k \|\mathbf{w}_t^k - \mathbf{w}_{\tau(t)}\|^2 I_{k,\tau(t)} \\
&\stackrel{(b)}{\leq} \mathbb{E} \sum_{k=1}^{N} p'_k I_{k,\tau(t)} \mathbb{E}_{\cdot|p'_k, I_{k,\tau(t)}} \left( \sum_{i=\tau(t)}^{\tau(t)+E-1} (E-1)\eta_i^2 \|\nabla F_k(\mathbf{w}_i^k)\|^2 \right) \\
&\stackrel{(c)}{\leq} \mathbb{E} \sum_{k=1}^{N} p'_k I_{k,\tau(t)} \sum_{i=\tau(t)}^{\tau(t)+E-1} (E-1)\eta_i^2 G^2 \\
&\stackrel{(d)}{\leq} \mathbb{E} \sum_{k=1}^{N} p'_k I_{k,\tau(t)} (E-1)^2 \eta_{\tau(t)}^2 G^2 \\
&\stackrel{(e)}{\leq} 4\eta_t^2 (E-1)^2 G^2,
\end{aligned} \tag{25}
$$

where $(a)$ is due to the definition $\mathbf{w}_t$ and the fact that (for vectors $\mathbf{a}_k$)

$$\sum_{k=1}^{N} \|\mathbf{a}_k - \bar{\mathbf{a}}\|^2 \leq \sum_{k=1}^{N} \|\mathbf{a}_k\|^2,$$

where $\bar{\mathbf{a}}$ is a weighted average of the $\mathbf{a}_k$'s, $(b)$ is due to the fact that when $t - \tau(t) \leq E - 1$

$$
\begin{aligned}
\|\mathbf{w}_t^k - \mathbf{w}_{\tau(t)}\|^2 = \left\| \sum_{i=\tau(t)}^{t} \nabla F(\mathbf{w}_i^k) \right\|^2 &\leq \sum_{i=\tau(t)}^{t} (E-1)\eta_i^2 \|\nabla F_k(\mathbf{w}_i^k)\|^2 \\
&\leq \sum_{i=\tau(t)}^{\tau(t)+E-1} (E-1)\eta_i^2 \|\nabla F_k(\mathbf{w}_i^k)\|^2,
\end{aligned}
$$

$(c)$ is due to Assumption 4, $(d)$ is because $\eta_t$ is non-increasing and $(e)$ is because $\eta_{\tau(t)} \leq 2\eta_t$. $\qquad\square$

**Lemma A.5.** *(SGD variance bound) Given Assumption 3,*

$$\mathbb{E}\|\bar{\mathbf{g}}_t - \mathbf{g}_t\|^2 \leq \sigma^2. \tag{26}$$

*Proof.* First we notice that

$$
\begin{aligned}
\mathbb{E}\|\bar{\mathbf{g}}_t - \mathbf{g}_t\|^2 &= \mathbb{E} \left\| \sum_{k=1}^{N} p'_k I_{k,\tau(t)} (\nabla F(\mathbf{w}_t^k; \xi_t^k) - \nabla F(\mathbf{w}_t^k)) \right\|^2 \\
&\leq \mathbb{E} \sum_{k=1}^{N} {p'_k}^2 I_{k,\tau(t)} \mathbb{E}_{\cdot|p'_k, I_{k,\tau(t)}} \|\nabla F(\mathbf{w}_t^k; \xi_t^k) - \nabla F(\mathbf{w}_t^k)\|^2 \\
&\stackrel{(a)}{\leq} \sigma^2 \mathbb{E} \sum_{k=1}^{N} {p'_k}^2 I_{k,\tau(t)} \stackrel{(b)}{\leq} \sigma^2,
\end{aligned} \tag{27}
$$

where $(a)$ is due to Assumption 3 and $(b)$ is because $p'_k \leq 1$ and $\sum_{k=1}^{N} p'_k I_{k,\tau(t)} = 1$.  $\square$

Notice that in Lemma A.5, if one instead has individual variance bounds of $\sigma_k^2$, one can also extend this lemma to obtain

$$\mathbb{E}\|\overline{\mathbf{g}}_t - \mathbf{g}_t\|^2 \leq \mathbb{E}\sum_{k=1}^{N} {p'_k}^2 \sigma_k^2 \tag{28}$$

in a straightforward manner. If required, the subsequent analysis can be easily extended to this case.

## A.2 One-step SGD

**Lemma A.6.** *(One step SGD) Assume Assumptions 1, 2, 3 and 4. If $\eta_t \leq \frac{1}{4L}$ and $\mathbf{w}^*$ is the global optimum of objective $F(.)$, then*

$$\mathbb{E}\|\mathbf{w}_{t+1} - \mathbf{w}^*\|^2 \leq (1 - \eta_t\mu)\mathbb{E}\|\mathbf{w}_t - \mathbf{w}^*\|^2 + \eta_t^2 B_t + \eta_t D_t, \tag{29}$$

*where*

$$
\begin{aligned}
B_t &:= 8(E-1)^2 G^2 + 6L\Gamma\mathbb{E}\left[\left(\frac{1}{1 + \sum_{k\notin\mathcal{P}} p_k I_{k,\tau(t)}}\right)\right] + \sigma^2 \\
D_t &= 2\mathbb{E}\left[\frac{\sum_{k\notin P} p_k I_{k,\tau(t)}\Gamma_k}{1 + \sum_{k\notin\mathcal{P}} p_k I_{k,\tau(t)}}\right] \\
\Gamma &:= F^* - \sum_{k\in\mathcal{P}} p_k F_k^*, \Gamma_k := F_k(\mathbf{w}^*) - F_k^*. 
\end{aligned}
\tag{30}
$$

We prove this lemma in the next section. Now define $\Delta_t = \mathbb{E}\|\mathbf{w}_t - \mathbf{w}^*\|^2$. We assume a diminishing step-size $\eta_t = \frac{2}{\mu(t+\gamma)}$, with $\eta_1 \leq \min\left(\frac{1}{4L}, \frac{1}{\mu}\right)$ and $\eta_t \leq 2\eta_{t+E}$ (The choice of $\gamma$ guarantees this condition). We can prove that $\Delta_t \leq \frac{v_t}{\gamma+t}$ by induction, where $v_t = \left(\frac{4}{\mu^2}\overline{B}_t + \frac{2}{\mu}\overline{D}_t(t+\gamma) + \gamma\Delta_0\right)$, where (if we pick $\gamma = \max\{\frac{8L}{\mu}, E\}$)

$$\overline{B}_0, \overline{B}_1 = 0, \overline{B}_t = \frac{1}{t+\gamma-2}\sum_{i=1}^{t-1} B_i \quad \forall t \geq 2,$$

$$\overline{D}_0, \overline{D}_1 = 0, \overline{D}_t = \frac{1}{t+\gamma-2}\sum_{i=1}^{t-1} D_t \quad \forall t \geq 2. \tag{31}$$

We proceed via induction on the number of time steps $t$.

**Base case.** When $t = 0$,

$$\Delta_0 \leq \frac{v_0}{\gamma} = \Delta_0, \tag{32}$$

from the definition of $v_t$.

**Inductive step.** Assume $\Delta_t \leq \frac{v_t}{\gamma+t}$. Then for $t+1$, due to Lemma A.6

$$
\begin{aligned}
\Delta_{t+1} &\leq \left(1 - \frac{2}{t+\gamma}\right)\Delta_t + \frac{4}{\mu^2(t+\gamma)^2}B_t + \frac{2}{\mu(t+\gamma)}D_t \\
&\overset{(a)}{\leq} \left(1 - \frac{2}{t+\gamma}\right)\frac{v_t}{t+\gamma} + \frac{4}{\mu^2(t+\gamma)^2}B_t + \frac{2}{\mu(t+\gamma)}D_t \\
&= \frac{4}{\mu^2(t+\gamma)^2}\left((t+\gamma-2)\overline{B}_t + B_t\right) + \frac{2}{\mu(t+\gamma)}\left((t+\gamma-2)\overline{D}_t + D_t\right) + \gamma\frac{(t+\gamma-2)\Delta_0}{(t+\gamma)^2} \\
&= \frac{4}{\mu^2(t+\gamma)^2}\left(\sum_{i=1}^{t-1} B_i + B_t\right) + \frac{2}{\mu(t+\gamma)}\left(\sum_{i=1}^{t-1} D_i + D_t\right) + \gamma\frac{(t+\gamma-2)\Delta_0}{(t+\gamma)^2} \\
&\overset{(b)}{\leq} \frac{4(t+\gamma-1)}{\mu^2(t+\gamma)^2}\overline{B}_{t+1} + \frac{2(t+\gamma-1)}{\mu(t+\gamma)(t+\gamma+1)}\overline{D}_{t+1}(t+\gamma+1) + \gamma\frac{\Delta_0}{t+\gamma+1} \overset{(c)}{\leq} \frac{v_{t+1}}{\gamma+t+1},
\end{aligned}
\tag{33}
$$

where $(a)$ is due to the inductive hypothesis, $(b)$ and $(c)$ are because

$$
\frac{t+\gamma-2}{(t+\gamma)^2} \leq \frac{(t+\gamma-1)}{(t+\gamma)^2} = \frac{(t+\gamma)^2-1}{(t+\gamma)(t+\gamma+1)} \leq \frac{1}{t+\gamma+1},
\tag{34}
$$

Thus, by $L$-Lipschitz smoothness)

$$
\mathbb{E}[F(\mathbf{w}_t)] - F^* \leq \frac{L}{2}\frac{v_t}{t+\gamma} \leq (C_1 + C_2\theta_T\Gamma) + \rho_T,
\tag{35}
$$

which proves Theorem A.1.

## A.3 Proof of Lemma A.6

*Proof of Lemma A.6.* By definition $\mathbf{w}_{t+1} = \mathbf{w}_t - \eta_t\mathbf{g}_t$. Therefore we can write

$$
\begin{aligned}
\|\mathbf{w}_{t+1} - \mathbf{w}^*\|^2 &= \|\mathbf{w}_t - \eta_t\mathbf{g}_t - \mathbf{w}^* - \eta_t\overline{\mathbf{g}}_t + \eta_t\overline{\mathbf{g}}_t\|^2 \\
&= = \underbrace{\|\mathbf{w}_t - \mathbf{w}^* - \eta_t\overline{\mathbf{g}}_t\|^2}_{A_1} + \underbrace{2\langle\mathbf{w}_t - \mathbf{w}^* - \eta_t\overline{\mathbf{g}}_t, \overline{\mathbf{g}}_t - \mathbf{g}_t\rangle}_{B} + \eta_t^2\|\mathbf{g}_t - \overline{\mathbf{g}}_t\|^2.
\end{aligned}
\tag{36}
$$

Notice that for $B$ in (36),

$$
\begin{aligned}
\mathbb{E}[B] &= \mathbb{E}[\langle\mathbf{w}_t - \mathbf{w}^* - \eta_t\overline{\mathbf{g}}_t, \overline{\mathbf{g}}_t - \mathbf{g}_t\rangle] \\
&= \langle\mathbf{w}_t - \mathbf{w}^* - \eta_t\overline{\mathbf{g}}_t, \overline{\mathbf{g}}_t - \mathbb{E}_\xi[\mathbf{g}_t]\rangle = 0,
\end{aligned}
\tag{37}
$$

where $E_\xi$, is the expectation over the distribution $\xi = \{\xi_1, \ldots, \xi_k\}$, the data sampled in a mini-batch by each client at time step $t$.

Now consider $A_1$ in (36). We can further split it into 3 terms as

$$
\|\mathbf{w}_t - \eta_t\overline{\mathbf{g}}_t - \mathbf{w}^*\|^2 = \|\mathbf{w}_t - \mathbf{w}^*\|^2 - \underbrace{2\eta_t\langle\mathbf{w}_t - \mathbf{w}^*, \overline{\mathbf{g}}_t\rangle}_{A_2} + \underbrace{\eta_t^2\|\overline{\mathbf{g}}_t\|^2}_{A_3}.
\tag{38}
$$

One can upper bound $A_3$ using Lemma A.2 ($L$-smoothness), which yields

$$
\eta_t^2\|\overline{\mathbf{g}}_t\|^2 \leq \eta_t^2\sum_{k=1}^{N} p_k'\|\nabla F_k(\mathbf{w}_t^k)\|^2 I_{k,\tau(t)} \leq 2L\eta_t^2\sum_{k=1}^{N} p_k'\left(F_k(\mathbf{w}_t^k) - F_k^*\right)I_{k,\tau(t)}.
\tag{39}
$$

Moreover considering $A_2$, we can expand it as

$$
-2\eta_t\langle\mathbf{w}_t - \mathbf{w}^*, \overline{\mathbf{g}}_t\rangle = -2\eta_t\sum_{k=1}^{N} p_k'\langle\mathbf{w}_t - \mathbf{w}^*, \nabla F_k(\mathbf{w}_t^k)\rangle I_{k,\tau(t)}.
\tag{40}
$$

Now each term $-2\eta_t\langle \mathbf{w}_t - \mathbf{w}^*, \nabla F_k(\mathbf{w}_t^k)\rangle$ can be upper bounded as

$$
-2\langle \mathbf{w}_t - \mathbf{w}^*, \nabla F_k(\mathbf{w}_t^k)\rangle = -2\langle \mathbf{w}_t - \mathbf{w}_t^k, \nabla F_k(\mathbf{w}_t^k)\rangle - 2\langle \mathbf{w}_t^k - \mathbf{w}^*, \nabla F_k(\mathbf{w}_t^k)\rangle
$$

$$
\leq \underbrace{\frac{1}{\eta_t}\|\mathbf{w}_t - \mathbf{w}_t^k\|^2 + \eta_t\|\nabla F_k(\mathbf{w}_t^k)\|^2}_{\text{AM-GM inequality}} \underbrace{- 2(F_k(\mathbf{w}_t^k) - F_k(\mathbf{w}^*)) - \mu\|\mathbf{w}_t^k - \mathbf{w}^*\|^2}_{\text{Strong Convexity}}, \tag{41}
$$

where the AM-GM inequality refers to the fact that the arthiemtic mean is greater geometric mean. Precisely for two vectors $\mathbf{z}_1, \mathbf{z}_2$, we have

$$
2\langle \mathbf{z}_1, \mathbf{z}_2\rangle = 2\langle a\mathbf{z}_1, (1/a)\mathbf{z}_2\rangle \leq a^2\|\mathbf{z}_1\|^2 + \frac{1}{a^2}\|\mathbf{z}_2\|^2, \tag{42}
$$

where setting $a = \sqrt{\eta_t}$, $\mathbf{z}_1 = \mathbf{w}^* - \mathbf{w}_t$ and $\mathbf{z}_2 = \nabla F_k(\mathbf{w}_t^k)$, leads to the desired inequality, and Strong Convexity is due to Assumption 2.

Combining (38), (39), (40) and (41) we obtain

$$
\|\mathbf{w}_{t+1} - \eta_t \overline{\mathbf{g}}_t - \mathbf{w}^*\|^2 \leq \|\mathbf{w}_t - \mathbf{w}^*\|^2 + 2L\eta_t^2 \sum_{k=1}^{N} p_k'(F_k(\mathbf{w}_t^k) - F_k^*)I_{k,\tau(t)}
$$

$$
+ \eta_t \sum_{k=1}^{N} p_k' \left(\frac{1}{\eta_t}\|\mathbf{w}_t - \mathbf{w}_t^k\|^2 + \eta_t\|\nabla F_k(\mathbf{w}_t^k)\|^2 - 2(F_k(\mathbf{w}_t^k) - F_k(\mathbf{w}^*)) - \mu\|\mathbf{w}_t^k - \mathbf{w}^*\|^2\right) I_{k,\tau(t)}
$$

$$
\overset{(a)}{\leq} \|\mathbf{w}_t - \mathbf{w}^*\|^2 - \mu\eta_t \sum_{k=1}^{N} p_k'\|\mathbf{w}_t^k - \mathbf{w}^*\|^2 I_{k,\tau(t)} + \sum_{k=1}^{N} p_k'\|\mathbf{w}_t - \mathbf{w}_t^k\|^2 I_{k,\tau(t)}
$$

$$
+ 4L\eta_t^2 \sum_{k=1}^{N} p_k'(F_k(\mathbf{w}_t^k) - F_k^*)I_{k,\tau(t)} - 2\eta_t \sum_{k=1}^{N} p_k'\left(F_k(\mathbf{w}_t^k) - F_k(\mathbf{w}^*)\right) I_{k,\tau(t)}
$$

$$
\overset{(b)}{\leq} (1 - \eta_t\mu)\|\mathbf{w}_t - \mathbf{w}^*\|^2 + \sum_{k=1}^{N} p_k'\|\mathbf{w}_t - \mathbf{w}_t^k\|^2 I_{k,\tau(t)}
$$

$$
+ \underbrace{4L\eta_t^2 \sum_{k\in\mathcal{P}} p_k'(F_k(\mathbf{w}_t^k) - F_k^*) - 2\eta_t \sum_{k\in\mathcal{P}} p_k'\left(F_k(\mathbf{w}_t^k) - F_k(\mathbf{w}^*)\right)}_{A_4}
$$

$$
+ \underbrace{4L\eta_t^2 \sum_{k\notin\mathcal{P}} p_k'(F_k(\mathbf{w}_t^k) - F_k^*)I_{k,\tau(t)} - 2\eta_t \sum_{k\notin\mathcal{P}} p_k'\left(F_k(\mathbf{w}_t^k) - F_k(\mathbf{w}^*)\right) I_{k,\tau(t)}}_{A_5}, \tag{43}
$$

where $(a)$ is due to (39) and $(b)$ follows from Lemma A.3. $A_4$ captures the aggregation of priority clients, while $A_5$ captures the aggregation of non-priority clients. We will bound them separately. First let us consider $A_4$ in (43) (recall $\eta_t \leq \frac{1}{4L}$)

$$
4L\eta_t^2 \sum_{k\in\mathcal{P}} p_k'(F_k(\mathbf{w}_t^k) - F_k^*) - 2\eta_t \sum_{k\in\mathcal{P}} p_k'\left(F_k(\mathbf{w}_t^k) - F_k(\mathbf{w}^*)\right)
$$

$$
= -\gamma_t \sum_{k\in\mathcal{P}} p_k'(F_k(\mathbf{w}_t^k) - F_k^*) + 2\eta_t \sum_{k\in\mathcal{P}} p_k'(F_k(\mathbf{w}^*) - F_k^*)
$$

$$
= -\gamma_t \sum_{k\in\mathcal{P}} p_k'(F_k(\mathbf{w}_t^k) - F^*) + 4L\eta_t^2 \sum_{k\in\mathcal{P}} p_k'(F^* - F_k^*)
$$

$$
= \underbrace{-\gamma_t \sum_{k\in\mathcal{P}} p_k'(F_k(\mathbf{w}_t^k) - F^*)}_{A_6} + 4L\eta_t^2 E_t, \tag{44}
$$

where $\gamma_t := 2\eta_t(1 - 2L\eta_t)$ and

$$E_t := \mathbb{E}\left[\frac{\Gamma}{1 + \sum_{k\notin\mathcal{P}} p_k I_{k,\tau(t)}}\right].$$

Now to upper bound $A_6$ in (44) we consider the following chain of inequalities

$$\sum_{k\in\mathcal{P}} p_k'(F_k(\mathbf{w}_t^k) - F^*) = \sum_{k\in\mathcal{P}} p_k'(F_k(\mathbf{w}_t^k) - F_k(\mathbf{w}_t)) + \sum_{k\in\mathcal{P}} p_k'(F_k(\mathbf{w}_t) - F^*)$$

$$\geq \sum_{k\in\mathcal{P}} p_k'\langle\nabla F_k(\mathbf{w}_t), \mathbf{w}_t^k - \mathbf{w}_t\rangle + \sum_{k\in\mathcal{P}} p_k'(F_k(\mathbf{w}_t) - F^*)$$

$$\overset{(a)}{\geq} -\frac{1}{2}\sum_{k\in\mathcal{P}} p_k'\left[\eta_t\|\nabla F_k(\mathbf{w}_t)\|^2 + \frac{1}{\eta_t}\|\mathbf{w}_t^k - \mathbf{w}_t\|^2\right] + \sum_{k\in\mathcal{P}} p_k'(F_k(\mathbf{w}_t) - F^*)$$

$$\overset{(b)}{\geq} -\sum_{k\in\mathcal{P}} p_k'\left[\eta_t L(F_k(\mathbf{w}_t) - F_k^*) + \frac{1}{\eta_t}\|\mathbf{w}_t^k - \mathbf{w}_t\|^2\right] + \sum_{k\in\mathcal{P}} p_k'(F_k(\mathbf{w}_t) - F^*), \tag{45}$$

where $(a)$ is due to the AM-GM inequality and $(b)$ is due to Lemma A.2. Using this, $A_6$ in (44) can be upper bounded as

$$A_6 \leq \gamma_t \sum_{k\in\mathcal{P}} p_k'\left[\eta_t L(F_k(\mathbf{w}_t) - F_k^*) + \frac{1}{2\eta_t}\|\mathbf{w}_t^k - \mathbf{w}_t\|^2\right] - \gamma_t \sum_{k\in\mathcal{P}} p_k'(F_k(\mathbf{w}_t) - F^*)$$

$$= \gamma_t(\eta_t L - 1)\sum_{k\in\mathcal{P}} p_k'(F(\mathbf{w}_t) - F^*) + \gamma_t\eta_t LE_t + \frac{\gamma_t}{2\eta_t}\sum_{k\in\mathcal{P}} p_k'\|\mathbf{w}_t^k - \mathbf{w}_t\|^2$$

$$\overset{(a)}{\leq} \gamma_t\eta_t LE_t + \frac{\gamma_t}{2\eta_t}\sum_{k=1}^N p_k'\|\mathbf{w}_t^k - \mathbf{w}_t\|^2 \overset{(b)}{\leq} \sum_{k\in\mathcal{P}} p_k'\|\mathbf{w}_t^k - \mathbf{w}_t\|^2 + \gamma_t\eta_t LE_t, \tag{46}$$

where $(a)$ is because $(F(\mathbf{w}_t) - F^*) \geq 0$ and $\gamma_t(\eta_t L - 1) \leq 0$, since we assume $\eta_L \leq \frac{1}{4L}$ and $(b)$ is because $\frac{\gamma_t}{2\eta_t} \leq 1$. Therefore (44) ($A_4$ in (41)) can be upper bounded as

$$A_4 \leq \sum_{k\in\mathcal{P}} p_k'\|\mathbf{w}_t^k - \mathbf{w}_t\|^2 I_{k,\tau(t)} + (\gamma_t\eta_t L + 4L\eta_t^2)E_t$$

$$\leq \sum_{k\in\mathcal{P}} p_k'\|\mathbf{w}_t^k - \mathbf{w}_t\|^2 I_{k,\tau(t)} + 6L\Gamma\eta_t^2\mathbb{E}\left[\frac{1}{1 + \sum_{k\notin\mathcal{P}} p_k I_{k,\tau(t)}}\right]. \tag{47}$$

Now consider $A_5$ in (43)

$$4L\eta_t^2 \sum_{k\notin\mathcal{P}} p_k'(F_k(\mathbf{w}_t^k) - F_k^*)I_{k,\tau(t)} - 2\eta_t \sum_{k\notin\mathcal{P}} p_k'\left(F_k(\mathbf{w}_t^k) - F_k(\mathbf{w}^*)\right)I_{k,\tau(t)}$$

$$= -\gamma_t \sum_{k\notin\mathcal{P}} p_k'(F_k(\mathbf{w}_t^k) - F_k^*)I_{k,\tau(t)} + 2\eta_t \sum_{k\notin\mathcal{P}} p_k'(F_k(\mathbf{w}^*) - F_k^*)I_{k,\tau(t)}$$

$$= \underbrace{-\gamma_t \sum_{k\notin\mathcal{P}} p_k'(F_k(\mathbf{w}_t^k) - F_k^*)I_{k,\tau(t)}}_{A_7} + \eta_t D_t. \tag{48}$$

where $\gamma_t := 2\eta_t(1 - 2L\eta_t)$ and (for $(\Gamma_k := F_k(\mathbf{w}^*) - F_k^*)$)

$$D_t := 2\mathbb{E}\left[\frac{\sum_{k\notin P} p_k\Gamma_k I_{k,\tau(t)}}{1 + \sum_{k\notin\mathcal{P}} p_k I_{k,\tau(t)}}\right]. \tag{49}$$

Now to upper bound to $A_7$ in (48), we consider the following chain of inequalities

$$\sum_{k \notin \mathcal{P}} p_k'(F_k(\mathbf{w}_t^k) - F_k^*)I_{k,\tau(t)} = \sum_{k \notin \mathcal{P}} p_k'(F_k(\mathbf{w}_t^k) - F_k(\mathbf{w}_t))I_{k,\tau(t)} + \sum_{k \notin \mathcal{P}} p_k'(F_k(\mathbf{w}_t) - F_k^*)I_{k,\tau(t)}$$

$$\geq \sum_{k \notin \mathcal{P}} p_k'\langle \nabla F_k(\mathbf{w}_t), \mathbf{w}_t^k - \mathbf{w}_t \rangle I_{k,\tau(t)} + \sum_{k \notin \mathcal{P}} p_k'(F_k(\mathbf{w}_t) - F_k^*)I_{k,\tau(t)}$$

$$\overset{(a)}{\geq} -\frac{1}{2}\sum_{k \notin \mathcal{P}} p_k'\left[\eta_t\|\nabla F_k(\mathbf{w}_t)\|^2 + \frac{1}{\eta_t}\|\mathbf{w}_t^k - \mathbf{w}_t\|^2\right]I_{k,\tau(t)} + \sum_{k \notin \mathcal{P}} p_k'(F_k(\mathbf{w}_t) - F_k^*)I_{k,\tau(t)}$$

$$\overset{(b)}{\geq} -\sum_{k \notin \mathcal{P}} p_k'\left[\eta_t L(F_k(\mathbf{w}_t) - F_k^*) + \frac{1}{\eta_t}\|\mathbf{w}_t^k - \mathbf{w}_t\|^2\right]I_{k,\tau(t)} + \sum_{k \notin \mathcal{P}} p_k'(F_k(\mathbf{w}_t) - F_k^*))I_{k,\tau(t)}, \qquad (50)$$

where $(a)$ is due to the AM-GM inequality and $(b)$ is due to Lemma A.2. Using this, $A_7$ in (48) can be upper bounded as

$$A_7 \leq \gamma_t \sum_{k \notin \mathcal{P}} p_k'\left[\eta_t L(F_k(\mathbf{w}_t) - F_k^*) + \frac{1}{2\eta_t}\|\mathbf{w}_t^k - \mathbf{w}_t\|^2\right]I_{k,\tau(t)} - \gamma_t \sum_{k \notin \mathcal{P}} p_k'(F_k(\mathbf{w}_t) - F_k^*)I_{k,\tau(t)}$$

$$= \gamma_t(\eta_t L - 1)\sum_{k \notin \mathcal{P}} p_k'(F_k(\mathbf{w}_t) - F_k^*))I_{k,\tau(t)} + \frac{\gamma_t}{2\eta_t}\sum_{k \notin \mathcal{P}} p_k'\|\mathbf{w}_t^k - \mathbf{w}_t\|^2 I_{k,\tau(t)}$$

$$\overset{(a)}{\leq} \frac{\gamma_t}{2\eta_t}\sum_{\notin \mathcal{P}} p_k'\|\mathbf{w}_t^k - \mathbf{w}_t\|^2 I_{k,\tau(t)} \overset{(b)}{\leq} \sum_{k \notin \mathcal{P}} p_k'\|\mathbf{w}_t^k - \mathbf{w}_t\|^2 I_{k,\tau(t)}, \qquad (51)$$

where $(a)$ is because for $\left(\eta_t \leq \frac{1}{4L}\right)$ and $\gamma_t(\eta_t L - 1) \leq 0$ and $(F_k(\mathbf{w}_t) - F_k^*) \geq 0$, and $(b)$ is again due $\frac{\gamma_t}{2\eta_t} \leq 1$. Therefore (48) ($A_5$ in (43)) can be upper bounded as

$$A_5 \leq \sum_{k \notin \mathcal{P}} p_k'\|\mathbf{w}_t^k - \mathbf{w}_t\|^2 I_{k,\tau(t)} + \eta_t D_t. \qquad (52)$$

Combining (36),(37),(43), (47) and (52), followed by taking expectation on both sides and using Lemmas A.4 and A.5, we infer that

$$\Delta_{t+1} \leq (1 - \mu_t\eta_t)\Delta_t + 8\eta_t^2(E-1)^2 G^2 + \eta_t^2 \sigma^2 + 6L\eta_t^2 E_t + \eta_t D_t, \qquad (53)$$

which proves Lemma A.6.

### A.4  Extensions to partial participation of clients

In this section, we generalize our findings to scenarios where only a randomly selected subset of priority indices are involved at any given time instance (with or without replacement). Additionally, we consider the case where non-priority clients engage according to any arbitrary participation pattern in each communication round.

Initially, we focus on the arbitrary participation of non-priority clients. A critical insight from our research is that for the $k^{\text{th}}$ non-priority client, the determination of their participation or non-participation in the aggregation during a specific communication round is guided by the indicator random variable $I_{k,\tau(t)}$. This variable is a general selection rule, flexible enough to encompass all potential situations, whether the server selects the clients, the clients voluntarily participate, or both.

For a broader applicability to arbitrary client cases, our analysis can be conveniently extended by redefining $I_{k,\tau(t)}$ for a non-priority client as

$$I_{k,\tau(t)} := \mathbf{1}\{|F_k\left(\mathbf{w}_t^k\right) - F\left(\mathbf{w}_t^k\right)| < \epsilon_t\} \times \mathbf{1}\{\text{Client } k \text{ participates in } \tau(t)^{\text{th}} \text{ communication round}\}. \qquad (54)$$

Under this revised definition of $I_{k,\tau(t)}$, our results remain valid. We consider this revised definition throughout this subsection. In a particular case, when

$$\mathbb{E}[\mathbf{1}\{\text{Client } k \text{ participates in } \tau(t)^{\text{th}} \text{ communication round}\}] = p,$$

this scenario corresponds to a selection method where each non-priority client is chosen uniformly at random with a probability $p$. This flexible integration of non-priority clients aligns with realistic situations, wherein certain non-priority clients may be excluded in every communication round due to factors such as unreliability or associated high computation or communication costs.

Now let us consider the selection pattern with replacement considered in Li et al. (2020a), which corresponds to partial priority-client participation. We also follow the proof skeleton in Li et al. (2020a). Let the set $\mathcal{S}_t$ define the clients participating at time step $t$. We refer to the following sampling scheme as "sampling with replacement".

The server sets $\mathcal{S}_t$ by sampling (with replacement) an index $k \in \mathcal{P}$ corresponding to probabilities generated by the data fraction $p_k : k \in \mathcal{P}$, recalling that $\sum_{k \in \mathcal{P}} p_k = 1$. Therefore, $S_t$ is a multiset. If $t$ corresponds to a communication round then the server aggregates the parameters as

$$\mathbf{w}'_{t+1} = \frac{1}{K} \sum_{k \in \mathcal{S}_{t+1}} \frac{\mathbf{w}^k_t}{1 + \sum_{k \notin \mathcal{P}} p_k I_{k,\tau(t)}} + \sum_{k \notin \mathcal{P}} p'_k \mathbf{w}^k_t. \tag{55}$$

The main insight to why we can prove our result is that this aggregation $\mathbf{w}'_{t+1}$ is an unbiased estimate of $\mathbf{w}_{t+1}$ defined in the previous section. Specifically under $\mathbb{E}_{\mathcal{S}_t | I_{k,\tau(t)}}$, the following lemmas (restated from Li et al. (2020a)) hold true.

**Lemma A.7.** *(Unbiased sampling scheme) If $t$ corresponds to a communication round, , we have*

$$\mathbb{E}_{\mathcal{S}_t | I_{k,\tau(t)}}[\mathbf{w}'_t] = \mathbf{w}_t.$$

**Lemma A.8.** *(Bounding the variance of $\mathbf{w}'_t$) Assume that Assumption 4 is true and $\eta_{\tau(t)} \leq 2\eta_t$ for all $t \geq 0$. If $t$ corresponds to a communication round then the expected variaance between $\mathbf{w}'_t$ and $\mathbf{w}_t$ is bounded by*

$$\mathbb{E}_{\mathcal{S}_t | I_{k,\tau(t)}} \|\mathbf{w}'_t - \mathbf{w}_t\|^2 \leq \frac{4\eta_t^2 E^2 G^2}{K}. \tag{56}$$

These lemmas are straightforward extensions to Lemmas 4 and 5 in Li et al. (2020a), so we skip the proof here.

Now consider one-step SGD, under this new aggregation method

$$
\begin{aligned}
\|\mathbf{w}'_{t+1} - \mathbf{w}^*\|^2 &= \|\mathbf{w}'_{t+1} - \mathbf{w}_{t+1} + \mathbf{w}_{t+1} - \mathbf{w}^*\|^2 \\
&= \|\mathbf{w}_{t+1} - \mathbf{w}^*\|^2 + 2\langle \mathbf{w}'_{t+1} - \mathbf{w}_{t+1}, \mathbf{w}_{t+1} - \mathbf{w}^* \rangle + \|\mathbf{w}'_{t+1} - \mathbf{w}_{t+1}\|^2 \\
&\overset{(a)}{\leq} \underbrace{\|\mathbf{w}_{t+1} - \mathbf{w}^*\|^2}_{B_1} + \frac{4\eta_t^2 E^2 G^2}{K},
\end{aligned}
\tag{57}
$$

where $(a)$ is due to Lemma A.7 and A.8. Notice that $B_1$ can be upper bounded with Lemma A.6. Therefore we get a upper bound with an added term (if we re-define $\Delta_t := \|\mathbf{w}'_t - \mathbf{w}^*\|^2$)

$$
\begin{aligned}
\Delta_{t+1} &\leq (1 - \mu_t \eta_t)\|\mathbf{w}_t - \mathbf{w}^*\|^2 + 8\eta_t^2 (E-1)^2 G^2 + \frac{4\eta_t^2 E^2 G^2}{K} + \eta_t^2 \sigma^2 + 6L\eta_t^2 E_t + \eta_t D_t \\
&\leq (1 - \mu_t \eta_t)\Delta_t + 8\eta_t^2 (E-1)^2 G^2 + \frac{8\eta_t^2 E^2 G^2}{K} + \eta_t^2 \sigma^2 + 6L\eta_t^2 E_t + \eta_t D_t.
\end{aligned}
\tag{58}
$$

The above equation (58) is similar to the bound in Lemma A.6, except for an additional bias due to the variance between $\mathbf{w}'_t$ and $\mathbf{w}_t$. Therefore we can use the same inductive arguments here too. We skip this since this is a straightforward extension of the proofs considered in the previous sections and state the a combined theorem, under partial participation of the priority clients and arbitrary participation of the non priority clients below.

**Theorem A.9.** *(Convergence under partial participation) Suppose that Assumptions 1, 2, 3, and 4 hold, and we set a decaying learning rate $\eta_t = \frac{2}{\mu(t+\gamma)}$, where $\gamma = \frac{8L}{\mu}$ and any $\epsilon_t \geq 0$. The expected error following $T$ total local iterations (equivalent to $T/E$ communication rounds) for* FedALIGN *under partial participation of the priority clients and arbitrary participation of the non-priority clients, satisfies*

$$\mathbb{E}[F(\mathbf{w}_T)] - F^* \leq \frac{1}{T+\gamma}\left(C_1' + C_2\theta_T\Gamma\right) + \rho_T, \tag{59}$$

*where $\theta_T \in [0,1]$ and $\rho_T$ can be adjusted by selecting an appropriate $\epsilon_t$ in each time step ($\rho_T$ and $\theta_T$ are defined as before). Here, $C_1'$ and $C_2$ are constants given by*

$$C_1' := \frac{2L}{\mu^2}\left(\sigma^2 + 8(E-1)^2G^2 + \frac{8E^2G^2}{K}\right) + \frac{4L^2}{\mu}\|\mathbf{w}_0 - \mathbf{w}^*\|^2, \quad C_2 := \frac{12L^2}{\mu^2}.$$

## B  Computational Complexity of FedALIGN

### Notation

Here we do a computational complexity analysis for FedALIGN.

- $N$: Total number of clients.

- $\mathcal{P}$: Set of participating clients in each round.

- $E$: Number of local epochs.

- $B$: Number of batches in the local dataset of each client.

- $t$: Communication round.

- $d$: Dimension of the model parameters.

- $\eta$: Learning rate.

### Server-Side Complexity

The server-side operations mainly involve broadcasting the model, aggregating updates, and checking conditions for client updates.

1. **Broadcasting the model parameters and loss $F(\mathbf{w}_t)$:**

   - Complexity: $O(N \cdot d)$ for broadcasting $\mathbf{w}_t$ and $F(\mathbf{w}_t)$.

2. **Aggregating updates from clients:**

   - Let $|\mathcal{P}|$ be the number of participating clients.
   - Complexity: $O(|\mathcal{P}| \cdot d)$ for summing the weighted updates.

3. **Computing the average model update:**

   - Complexity: $O(d)$ for dividing by the total weight $p$.

4. **Overall server-side complexity per round:**

$$O(N \cdot d + |\mathcal{P}| \cdot d + d) = O(N \cdot d)$$

**Client-Side Complexity**

The client-side operations involve updating the model using local data, which is where most of the computational effort is spent.

1. **Computing local loss $F_k(\mathbf{w}_t)$:**
   - Complexity: $O(B \cdot d)$, where $B$ is the number of batches.

2. **ClientUpdate function:**
   - For each epoch $E$:
     - For each batch $B$:
       * Complexity of gradient computation and model update: $O(B \cdot d)$.
   - Since this operation is nested within $E$ epochs, the complexity for each client update is:

$$O(E \cdot B \cdot d)$$

**Overall Complexity per Communication Round**

Combining the server and client operations, we get the overall complexity per communication round:

1. **Server-side operations:**
$$O(N \cdot d)$$

2. **Client-side operations:**
   - For participating clients ($|\mathcal{P}|$):

$$O(|\mathcal{P}| \cdot (E \cdot B \cdot d + B \cdot d)) = O(|\mathcal{P}| \cdot E \cdot B \cdot d)$$

3. **Combining both, the total complexity per communication round $t$ is:**

$$O(N \cdot d + |\mathcal{P}| \cdot E \cdot B \cdot d)$$

4. **In the worst case, where all clients participate ($|\mathcal{P}| = N$), the complexity is:**

$$O(N \cdot d + N \cdot E \cdot B \cdot d) = O(N \cdot E \cdot B \cdot d)$$

**Summary**

The computational complexity of the FedALIGN algorithm per communication round is dominated by the client-side updates and is given by:
$$O(N \cdot E \cdot B \cdot d)$$

where:

- $N$ is the number of clients,
- $E$ is the number of local epochs,
- $B$ is the number of batches in the local dataset,
- $d$ is the dimension of the model parameters.

## C   Experiment Details

We first describe the models and data distributions we used to run our main experiments in more detail. All our code is available in the supplementary material.

## C.1 Benchmark Datasets

Following the procedure in Brendan McMahan et al. (2017), we split the data into shards and assign multiple shards to each client. The loss used in all models is the cross-entropy loss.

For the FMNIST dataset we trained a Logistic Regression model with $784 \times 10$ fully connected layer. The dataset is split into 120 shards with each shard containing 500 samples of a single class. Each of the 60 clients are assigned two shards each.

For the *balanced*-EMNIST dataset we trained a 2-NN network with $784 \times 200$ input layer, followed by a $200 \times 200$ hidden layer and $200 \times 47$ output layer. The dataset is split into 600 shards with each shard containing 180 samples of a single class. The 60 clients are provided with 24 shards each.

For the CIFAR10 dataset we use a CNN with the first convolutional layer $5 \times 5$ kernel, with 32 output channels. The next convolutional layer again uses a $5 \times 5$ kernel with 64 output channels. We then attach sequential layers (fully connected) with $(512 \times 128)$ dimension. This is followed by a $128 \times 10$ output layer. The nodes with ReLU activation include a Kaiming initialization of weights. We also include batch normalization after the second convolutional layer and both the sequential layers. The data assignment is the same as that of FMNIST.

## C.2 Synthetic Datasets

We follow an extended version of the setup Li et al. (2018) for generating synthetic data. For each device $k$, samples $(X_k, Y_k)$ are generated according to the model $y = \mathrm{argmax}(\mathrm{softmax}(Wx + b))$, where $x \in R^{60}$, $W \in R^{10 \times 60}$, and $b \in R^{10}$. Heterogeneity is introduced by drawing $W_k$ and $b_k$ from normal distributions, $N(u_k, 1)$, with $u_k \sim N(0, \alpha)$. Each input $x_k$ is drawn from $N(v_k, \Sigma)$, where $\Sigma$ is a diagonal matrix with $\Sigma_{j,j} = j^{-1.2}$. Each element in the mean vector $v_k$ is drawn from $N(B_k, 1)$, where $B_k \sim N(0, \beta)$. For our main experiments we choose $\alpha = 1$, while $\beta = 1$. This gives us the data possessed by the priority clients.

We extend the setup to include non-priority clients where global data is distributed and progressively noisy data is added. Two forms of noise are considered: 1) Label flips, with the maximum noise range across all non-priority clients determined by "label noise factor" and "label noise skew factor" controls how skewed towards this maximum noise clients are. If "label noise skew factor", it means you have more high noise non-priority clients. 2) Irrelevant independent data points, with the maximum fraction of irrelevant data points determined by "random data fraction factor" and "random data fraction skew factor" controls how skewed towards containing this maximum clients are, in similar way as before.

These noise additions aim to model more realistic scenarios of heterogeneity between priority and non-priority clients, where non-priority clients are likely to contain down-sampled, low quality and irrelevant data that may harm the objective generated by the priority clients. We use the skews to model that these non-priority clients in general are aligned at variable levels to the global objective. High skews imply a larger number of non-priority clients are misaligned.

We use the SYNTH$(1, 1)$ for our experiments, which is generated by fixing $\alpha = 1$ and $\beta = 1$. In constructing our non-priority clients, we fix "random data fraction factor" $= 1$ and "label noise factor" $= 2.5$. We then pick three different parameter values for "label noise skew factor" $= 0.5, 1.5, 5$ and "random data fraction skew factor" $= 0.5, 1.5, 5$. These specific skew factors correspond to tags of low, medium and high noise in Figure 2.

# D  Additional Experiments

In this section we consider some additional experiments that further highlight the efficacy of FedALIGN.

**Outline of additional experiments.** In the following subsections we add the following additional experiments:

- Sensitivity of FedALIGN to varying $\epsilon$

- Performance of FedALIGN vs Models trained locally of Non-Priority Clients.

- Performance of FedALIGN vs Baselines, adapted for FedProx.

- Performance of FedALIGN vs Baselines under partial client participation.

- Sensitivity of FedALIGN to number of Priority Clients.

- Sensitivity if FedALIGN to number of warm-up rounds.

- More experiments under different number of priority clients, different number of local iterations and different noise skews for the synthetic data.

- Figure summarizing the sensitivity of FedALIGN (and other methods) to non-priority.

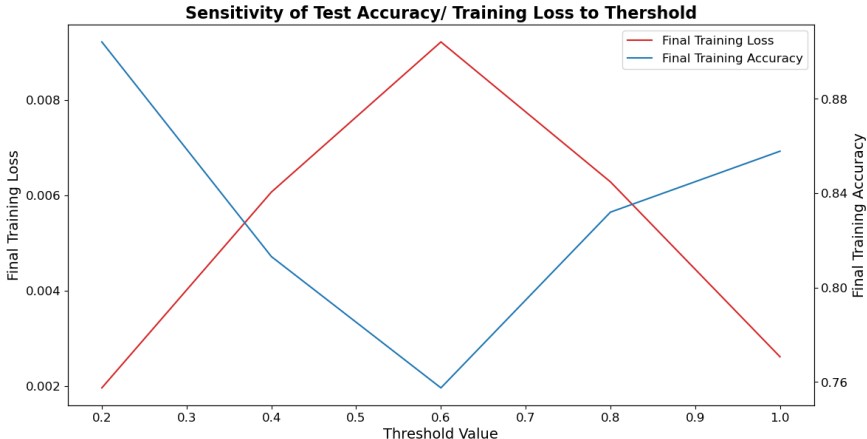

Figure 3: Sensitivity of FedALIGN to $\epsilon$ for FMNIST dataset. The graph shows the accuracy for different $\epsilon$ values, with $\epsilon = 0.6$ providing the best performance for a priority fraction of 0.1. The experiments were run for 50 communication rounds each.

### D.1   Sensitivity of FedALIGN to varying $\epsilon$

In our experiments to assess the sensitivity to $\epsilon$ for the FMNIST dataset, we found that $\epsilon = 0.6$ provides the best performance for a priority fraction of 0.1. Notably, $\epsilon = 0$ corresponds to no non-priority clients selected, while $\epsilon = 1$ corresponds to the every non-priority clients selected in each round approach. Figure 3 illustrates these results, demonstrating the impact of different $\epsilon$ values on model accuracy. These findings underscore the importance of selecting an optimal $\epsilon$ value to balance the trade-off between inclusion of non-priority clients and alignment, ultimately boosting the overall model performance. Our intuition supports this: including too few clients fails to add additional computation, while including too many dilutes adds a lot of bias, leading to degradation in performance. By fine-tuning $\epsilon$, we can strike the right balance.

### D.2   Performance of FedALIGN vs models trained locally at non-priority clients

In our experiments, we assess the accuracy of models trained using FedALIGN on test data of non-priority clients. To simulate a realistic, resource-constrained environment, we consider a scenario in which each client possesses only 50 samples (as opposed to 500 samples each client possessed in our main experiments). This scenario more closely models real-world situations where local models are hindered by limited computational power or inadequate data to develop an optimal model.

Our experimental findings, represented in Figure 4, demonstrate a significant performance enhancement for FedALIGN over local models across all our benchmark datasets in this setting. This result underlines a

compelling motivation for non-priority clients to participate in the federated learning framework, since they stand to benefit from a more robust global model.

However, as we have outlined in the main text, the superiority of a global model is not the sole reason incentivizing client participation. A working global model consistently aids in refining local models, which further boosts their performance. Thus, FedALIGN produces satisfactory models, which adds another layer of incentive for clients to join the federation.

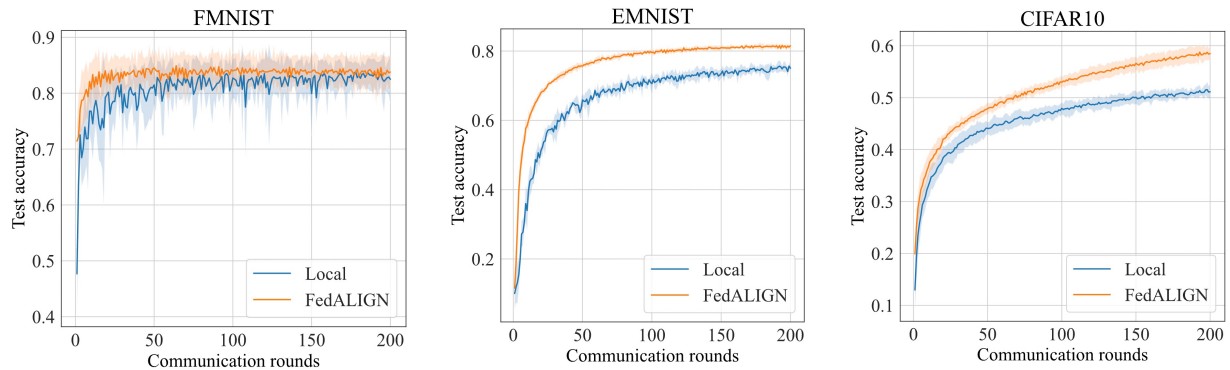

Figure 4: Test accuracy for benchmark datasets, on test data generated for local clients, where each client possess 50 data samples. FedALIGN (orange), converges faster and performs better than the local model.

### D.3   Performance of FedALIGN vs Baselines, adapted to FedProx

In our experiment, we consider FedProx Li et al. (2018) by incorporating a proximal regularization term (between the global and local model) in the local loss function of each client, amplified by a factor $\mu$ set at 1. We investigate a setting with 4 priority clients and 60 non-priority ones, maintaining the data distribution outlined in the paper's main section. The value of $\epsilon$ is set to 0.2.

We establish two baseline scenarios: one applying FedProx exclusively to the priority clients, the other extending it to all clients. These are compared against our FedALIGN variant, tailored to FedProx. Notably, our selection criteria operates as a separate, algorithm-independent step, simplifying its application to FedProx.

As illustrated in Figure 5, the results demonstrate FedALIGN's clear superiority over the baselines, underscoring its effectiveness in this context.

### D.4   Performance of FedALIGN vs Baselines under the partial participation scenario

In this experiment, we examine a partial participation scenario where subsets of clients, both priority and non-priority, are selected uniformly at random in each communication round. The client updates received from this subset form the basis of our investigation.

For this case, we consider a total of 60 clients, with 18 designated as priority clients. This increased count of priority clients is required for sufficient representation in every communication round. However, this configuration induces a more homogeneous global objective since more clients contributing to the global objective, encompasses more classes. For EMNIST, since each client gets 24 shards, we designate 4 clients to be priority clients (sufficient to lead to homogeneity).

As anticipated, we observe an improvement in FedALIGN performance under these circumstances. As predicted, the gains are not as significant as those we illustrated in the main result, where we only consider 2 priority clients, where the advantage of faster convergence is particularly evident because it is a more heterogeneous setup. The reason for this is because the full participation case considers 2 external clients with only 4 classes, which leads to significant heterogeneity. The reason for the smaller gain is due to the increased

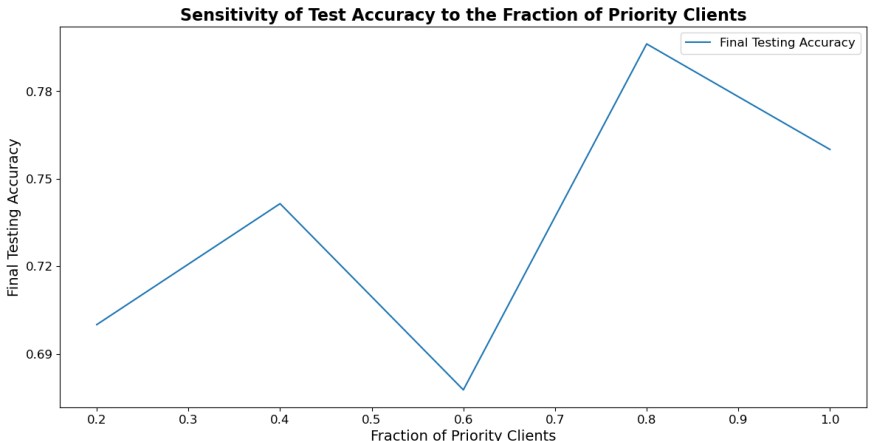

Figure 5: Test accuracy for FMNIST and EMNIST, when FedALIGN is adapted to FedPROX case, under full participation, with 4 priority clients and $N = 60$ clients, with $E = 5$. There is a clear advantage to using FedALIGN both in terms of convergence speed and improved test accuracy.

homogeneity in the global objective when more clients are included. Moreover we point to the synthetic experiments in Figure 2, which allow us to control heterogeneity between priority clients and how well aligned the non-priority clients are, even when we include more priority clients. This models more realistic cases of practical PFL and FedALIGN does significantly well in those cases. In subsection D.7, we demonstrate this phenomenon under the full participation case, further underscoring the importance of client heterogeneity being key to the success of FedALIGN, which is an important property in realistic FL settings.

For this experiment, we chose 0.3 (0.5 for the case of EMNIST, because there are lesser priority clients) as the fraction of clients involved in each round and set the value of $\epsilon$ to 0.2. Thus, while our setup performs efficiently even with partial participation, the experiment underscores the impact of client diversity on the effectiveness of FedALIGN.

Figure 6: The impact of varying the fraction of priority clients on model performance using FMNIST with $\epsilon = 0.2$. The x-axis represents the fraction of clients designated as priority, ranging from 0.2 to 1. We use 60 total clients. The y-axis shows the corresponding model accuracy. Initially, the accuracy degrades, likely because including more priority clients leads to more heterogeneity. Then the accuracy loss are counterbalanced by the more additional updates from priority clients, leading to a complex dependency and varying sensitivity.

### D.5 Sensitivity of FedALIGN to increasing fraction of priority clients.

In another experiment, we varied the fraction of clients designated as priority clients, ranging from 0.2 to 1, with 1 meaning all clients are included, using FMNIST with $\epsilon = 0.2$ (see Fig. 6).

We observed two competing factors influencing the model's performance. When the priority fraction is low, including more non-priority clients can increase accuracy due to the additional updates. However, this also introduces bias from the non-priority clients which reduces accuracy. Conversely, as we increase the number of priority clients, we achieve more aligned gradient updates but lose the benefit of extra updates from non-priority clients. This competition results in a complex dependency, causing sensitivity to vary significantly as the fraction of non-priority clients changes. .

### D.6 Sensitivity of FedALIGN to number of warm-up rounds.

We examined the sensitivity of FedALIGN to the number of warm-up rounds. We set a total of 50 communication rounds and varied the number of warm-up rounds from 0 to 40, using FMNIST with $\epsilon = 0.2$ (see Fig. 7).

We observed that initially, increasing the number of warm-up rounds improves performance, as it allows the model to better tune to the data from priority clients. However, as the number of warm-up rounds continues to increase, performance begins to decline. This is because the additional warm-up rounds come at the expense of updates from non-priority clients, leading to a trade-off between tuning to priority clients and incorporating diverse updates.

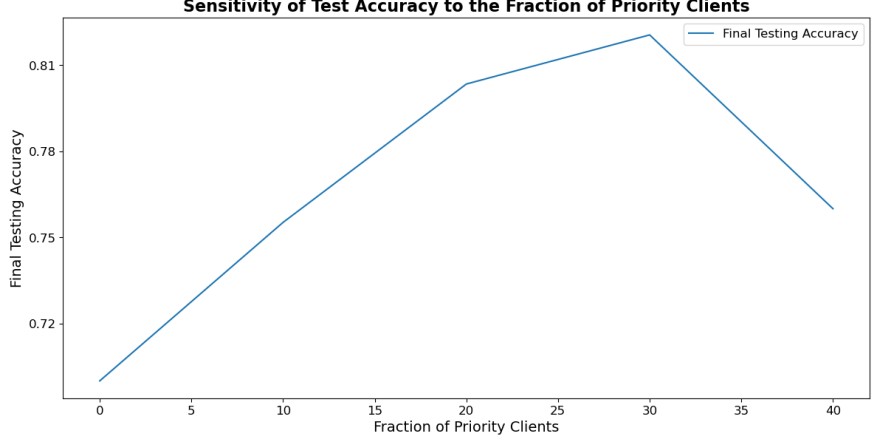

Figure 7: We set a total of 50 communication rounds and varied the number of warm-up rounds from 0 to 50, using FMNIST with $\epsilon = 0.2$. We observed that initially, increasing the number of warm-up rounds improves performance, as it allows the model to better tune to the data from priority clients. However, as the number of warm-up rounds continues to increase, performance begins to decline. This is because the additional warm-up rounds come at the expense of updates from non-priority clients, leading to a trade-off between tuning to priority clients and incorporating diverse updates.

### D.7 More experiments under different parameter choices

Here we provide results for additional experiments under various different parameter choices, where we can consider cases where more clients are included in the priority set and different local updates. Clearly as we start increasing the number of priority clients, the gap between the performance gains in FedALIGN is lower. This is due to a reduced heterogeneity in the global objective as you start increasing the number of clients. In the second plot of Figure 9 we also consider reducing the number of local updates to $E = 3$. We set $\epsilon = 0.2$ for all these experiments.

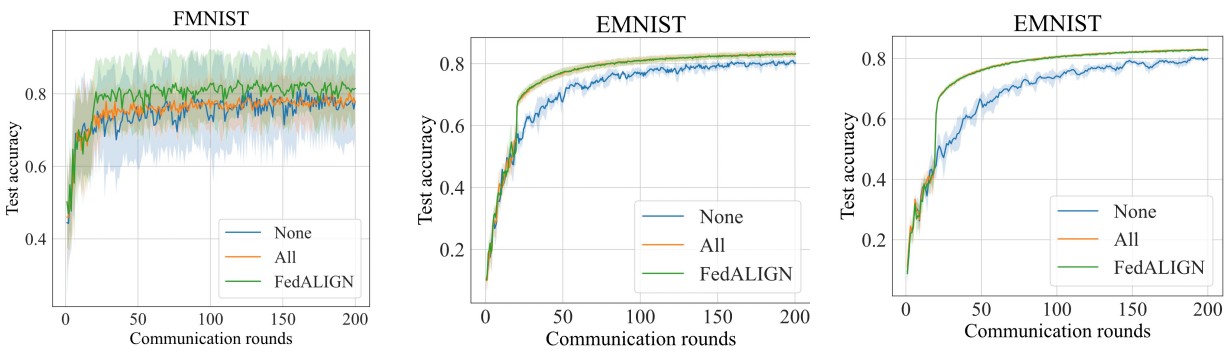

Figure 8: Test accuracy for benchmark datasets, in the partial participation case, with 18 priority clients and $N = 60$ clients for FMNIST and CIFAR, and 4 priority clients for EMNIST, with $E = 5$.

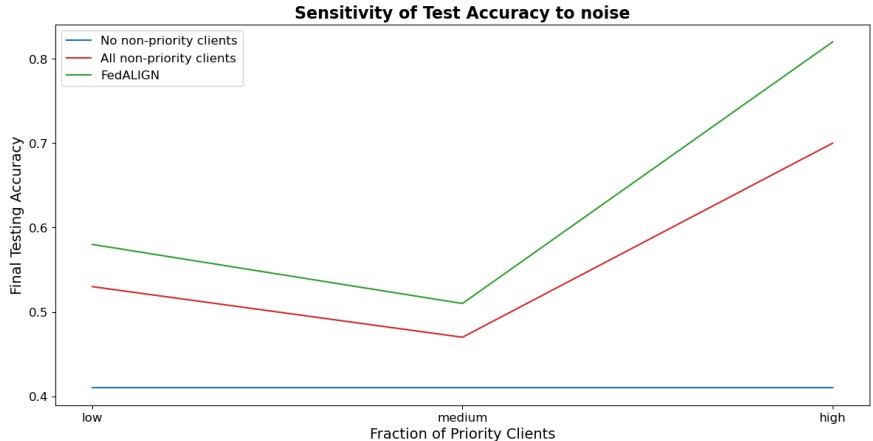

Figure 9: (1) Test accuracy for FMNIST, in the full participation case, with 6 priority clients and $N = 60$ clients, with $E = 5$, (2) Test accuracy for EMNIST, in the full participation case, with 9 priority clients and $N = 60$ clients, with $E = 3$ and (3) Test accuracy for EMNIST, in the full participation case, with 18 priority clients and $N = 60$ clients, with $E = 5$.

Figure 10: Summary of the results in Figure 2. FedALIGN does consistently better than other methods: None and FedAVG. None remains constant because the experiment is the same irrespective of non-priority client alignment