# OpenReview forum: "Prioritized Federated Learning: Leveraging Non-Priority Clients for Targeted Model Improvement"
_TMLR — Accepted by TMLR_

### Review · Reviewer_ufLW · 2024-05-30

**Summary Of Contributions:**

This paper introduces Prioritized Federated Learning (PFL), a variant of federated learning (FL) that optimizes the learning process by prioritizing certain clients. The proposed FedALIGN algorithm selects non-priority clients for inclusion based on the similarity of their data loss to the global model’s loss. This selection aims to enhance convergence rates and model accuracy by incorporating updates only from well-aligned non-priority clients. A theoretical analysis of FedALIGN is presented, demonstrating its potential for faster convergence with controlled bias, and these claims are validated through empirical results on synthetic and benchmark datasets. The findings suggest that FedALIGN effectively manages data heterogeneity.

**Audience:**

Yes

**Claims And Evidence:**

Yes

**Requested Changes:**

It is requested that the authors elaborate on the method used to determine the optimal choice of $\epsilon_t$ in their experiments. Additionally, please provide the computational complexity associated with this search process, which will help in assessing the practical applicability of FedALIGN in diverse settings.

**Strengths And Weaknesses:**

The problem setting addressed in this paper is compelling and relevant to real-world applications. The authors provide practical scenarios where the proposed algorithm could be beneficial. The main sections of the paper are well articulated and accessible, making the concepts and methodologies clear to the reader.

A significant concern is the choice of the threshold value $\epsilon_t$. The authors mention that a decreasing $\epsilon_t$ from 0.2 to 0 performed best in their experiments. However, this selection strategy may not generalize well across different datasets or conditions.

---

> ### Author Response · Authors · 2024-07-09
> **Response to reviewer ufLW**
>
> We appreciate the reviewer's request for elaboration on the method used to determine the optimal choice of epsilon_T in our experiments. In response to this, we have added an $\epsilon_T$ optimization analysis in Appendix D.1.
>
> Through our experiments, we observed that both decaying epsilon_T and keeping it constant yield similar results in practical settings. This is likely due to the heterogeneity stemming primarily from class imbalance rather than other factors.
>
> Furthermore, we have included a detailed computational complexity analysis in Appendix B, covering both client-side and server-side operations. This analysis will help in assessing the practical applicability of FedALIGN in diverse settings.
>
> Thank you for your insightful feedback, which has helped us enhance the clarity and thoroughness of our work.

---

### Review · Reviewer_TRig · 2024-06-11

**Summary Of Contributions:**

The submission proposes Prioritized Federated Learning (PFL), introducing a novel algorithm FedALIGN (Federated Adaptive Learning with Inclusion of Global Needs). The algorithm addresses the critical issue of client selection in federated learning, specifically focusing on leveraging non-priority clients to enhance model performance for priority clients.

1.Definition of PFL. The objective is to minimize the weighted mean objective of a subset of clients (priority clients).
2. Introduction of FedALIGN. which includes non-priority clients based on their alignment with the global objective, measured by the similarity of the model’s loss on their data compared to the global data.
3. FedALIGN employs a matching strategy, ensuring non-priority client gradients are utilized only when beneficial for priority clients.
4. Adaptive capacity to exploit well-aligned non-priority clients while preventing misaligned clients from negatively impacting the model.
5. Convergence analysis quantifying the trade-off between client selection and convergence speed.
6. Introduction of parameters \(\theta_T\) and \(\delta_T\) to balance the inclusion of non-priority clients and the resulting bias.

**Audience:**

Yes

**Broader Impact Concerns:**

NAN

**Claims And Evidence:**

Yes

**Requested Changes:**

### Requested Changes

1.  Provide a more comprehensive analysis of the proposed method's performance under highly heterogeneous data distributions. Include experiments and theoretical justifications for how FedALIGN handles scenarios with significant data heterogeneity among clients.

2: Extend the convergence analysis to account for partial participation scenarios. Include theoretical proofs and empirical results demonstrating the performance and convergence behavior of FedALIGN under varying degrees of client participation.

3: Propose and implement optimizations to reduce the computational and communication overhead of the alignment criteria. Provide scalability experiments with a large number of clients to demonstrate the improved efficiency and effectiveness of the proposed optimizations.

4: Clarify the intended behavior of the parameter epsilon_T. If epsilon_T is meant to decay to 0, update the experiments to reflect this decay process. Provide a detailed explanation and justification for the choice of all hyperparas including epsilon_T values used in the experiments.

6. Include experiments with more complex and diverse model architectures, such as deep neural networks (DNNs) and advanced convolutional neural networks (CNNs). Demonstrate the effectiveness of FedALIGN across a wider range of model complexities.

**Strengths And Weaknesses:**

1. Assumes the data distribution across clients is such that non-priority clients can be aligned with the global objective. This assumption is overly simplistic and fails to account for highly heterogeneous data distributions common in federated learning environments.
2. The bias term theta_T is not rigorously defined or justified. The introduction of this term can lead to a model that is heavily biased towards certain client data, reducing overall model performance. It makes difficult for balancing the trade-off between accelerated convergence and the bias introduced by including non-priority clients.
3. The convergence analysis is limited to full client participation scenarios, which is unrealistic. Partial participation, where only a subset of clients participates in each round, is more common and not adequately addressed.
4. The algorithm does not scale well with the number of clients. As the number of clients increases, the computational and communication overhead of maintaining and updating the alignment criteria grows significantly.
5. The epsilon_T was mentioned to be decayed to 0 but in the experiments it was set to be 0.2.
6. The model adopted were two layer CNN and LR, which was not sufficient to justify the effectiveness of the methods.

---

> ### Author Response · Authors · 2024-07-09
> **Response to Reviewer TRig**
>
> We thank the reviewer for their requested changes and reviews.
>
> **Requested change 1**
>
> We apologize for not being clear in the main body. We want to capture the realistic scenario for our “Prioritized” setting where there is a priority objective and non-priority clients with various levels of alignment. For benchmark data to impart class skews, we construct our experiments such that only certain classes exist within the dataset possessed by prioritized clients.The non-priority clients which now have 2 classes each will be well-aligned, if a group of them possess the same classes as priority clients and misaligned otherwise, leading to diversity amongst non-priority clients. This also leads to heterogeneity between the global objective and objective of each individual non-priority client, because only a few of them will be well-aligned. In traditional FL highly heterogeneous data distributions still contribute to the global objective. In PFL it does not, so it is justified to consider clients of various levels of alignment.
>
> **Requested change 2**
>
> We apologize for not making it clear in the main body. We do indeed consider the partial participation case in the Appendix. Specifically we do convergence analysis in Section A.4 and experiments in section D.4.
>
> **Requested change 3**
>
> We appreciate the reviewer's suggestion regarding the optimization of computational and communication overhead in the alignment criteria. We acknowledge that this is an important consideration, especially for scalability in practical applications with a large number of clients.
>
> However, our paper aims to introduce the concepts and setting of Prioritized Federated Learning (PFL). As such, it focuses primarily on laying the groundwork for understanding the basic principles and methodologies in this emerging field.
>
> While we recognize the significance of your suggestion, delving into the detailed optimizations and scalability experiments would extend beyond the scope of our current introductory work.
>
> **Requested Change 4**
>
> We appreciate the reviewer's request for clarification on the intended behavior of the parameter epsilon_T. In our experiments, epsilon_T was kept constant at 0.2. Through our analysis, we found that decaying epsilon_T to 0 yielded similar results to maintaining it at 0.2, primarily because the heterogeneity in our data stemmed mainly from class imbalance rather than other factors.
> To assist in optimizing the choice of epsilon_T, we have included sensitivity plots in Appendix D.1. These plots provide a comprehensive view of the impact of different epsilon_T values on the performance of our model. They serve as a valuable resource for fine-tuning epsilon_T in various scenarios.
>
> **Requested Change 5**
>
> We recognize the importance of demonstrating the effectiveness of FedALIGN across a wider range of model complexities.
> However, our current work aims to provide a foundational introduction to the setting of Prioritized Federated Learning (PFL). As such, we have focused on fundamental principles and methodologies to establish a baseline understanding. Including experiments with more complex models, while valuable, would extend beyond the scope of this introductory paper.
> We do, however, see the significance of your suggestion and plan to explore the effectiveness of FedALIGN with various model architectures in our future research endeavors.
>
> Thank you for your detailed feedback!

---

### Review · Reviewer_hCpB · 2024-06-26

**Summary Of Contributions:**

The paper presents a new framework for federated learning that targets a predetermined set of priority clients among a larger pool of clients. In this new framework, the authors propose a simple algorithm that can leverage the information of non-preferred clients to speed-up the convergence process. The idea is to incorporate non-preferred clients if they are well aligned with the set of priority clients. A theoretical analysis uncovers the trade-offs between increased convergence speed and additional bias introduced by non-preferred clients. Experiments demonstrate the benefits of the proposed approach on tailored problems.

**Audience:**

Yes

**Broader Impact Concerns:**

The introduction of preferred clients in federated learning may have broader economical impacts if adopted.
The authors argue that such a framework give more fairness. The viewpoint from a researcher in economy may help discuss such an impact.

**Claims And Evidence:**

Yes

**Requested Changes:**

See the main weaknesses part. The main requested changes are
- Comparison with prior work on client selection by finding a common ground to understand the benefits of each approach from scientific evidence.
- Analysis of sensibilities to the thresholds $\epsilon_t$ and to the proportion of preferred clients. Also maybe summarize analysis of sensibility to alignment of non-preferred clients (i.e. Fig. 2)
- Analysis to the sensitivity to the fraction of warmup steps. In particular an experiment without warm-up steps would be interesting.

I would also appreciate answers to the additional remarks and questions to the best of the authors abilities.

**Strengths And Weaknesses:**

**Strengths**
- The framework is interesting and could be leveraged to explore further research. For example, the authors mention several times the idea of ``incentivizing'' the clients which may be cast in a bandit framework.
- The algorithm is simple which ensures a simple integration in usual frameworks.
- The theoretical analysis provides meaningful practical insights on the implementation of the algorithm.
- The experiments demonstrate the benefits of the proposed approach on problems that satisfy the assumptions.

**Weaknesses**
- The paper will benefit from finding common ground with previous approaches to better understand its benefits. The authors insist that direct comparisons are not possible. I believe that the paper would actually benefit from finding common grounds to understand how well the algorithm performs according to some metrics compared to others. To give an example of such a presentation, in the context of heterogeneity, numerous distributionally robust methods have been designed. Such methods present tradeoffs between accuracy on the whole population and accuracy on the tail population. So in such papers both metrics (on the whole population and on the tail population) are presented. All methods can run but present different advantages.
- To give a full perspective on the approach, sensibilities to diverse hyperparameters should be evaluated and summarized. For example, sensibility to the thresholds $\epsilon_t$ need to be discussed. A plot of the final accuracy (or the area under the curve of the accuracy along time to capture convergence speed) versus the choice of epsilon would be helpful. Similarly, as pointed out in the Appendix, the benefits of the approach can vary dramatically with the proportion of priority clients. A plot summarizing such sensibility would be more helpful than a collection of optimization curves for different proportions. Same for the sensibility to noise: the experiments are interesting but could be summarized in a more meaningful way.

Overall the relevance of the approach may be questionable but it's ok for a publication at TMLR. Nevertheless a complete analysis of the approach with all its perks and pitfalls is necessary.

**Additional remarks and questions**
- Rather than coming up with their own definition of fairness with a lengthy argumentation (though an interesting one for philosophical discussion), the authors should consider appropriate references on fairness to strengthen their point.
- Eq. (5) typo $F_i$ should be $F_k$
- "While client selection as a concept is similar to our problem setting, the methods used for client
selection in classical settings often use updates from all the clients while making decisions, which is unrealistic
in the PFL setting. Therefore these algorithms do not serve as natural/direct baselines for our setting" -> I really don't understand why it would be unrealistic. The current algorithm also uses communications with all clients. The "unrealistic part" needs to be illustrated with a concrete experiment.
- In Algo 1:
  - Give names to the input parameters to facilitate its reading
  - I think there are mistakes about the definition of $p$. It's supposed to be a data fraction (so between 0 and 1). But it is inialized at 1 and then updated as $p \leftarrow p + p_k$ which makes it greater than 1.
- "We argue that the receipt of a well-performing global model provides a compelling incentive for non-priority clients to contribute to the aggregation" What is a compelling incentive? The right framework may be to consider incentives as utility functions (see literature from economy). One may then analyze formally what a ``compelling incentive'' is. Right now the associated paragraph lacks scientific ground.
- (Detail) "We posit", "We argue" (often in the text): A perk of working in applied mathematics is that arguments may be formalized and tested rigorously. Maybe consider putting comments without evidence in the conclusion. The points raised by the authors can be particularly relevant to build on the proposed approach
- Why do we need a warmup phase? What happens without the warmup phase? What is the impact of varying the fraction of warmup steps? (Also maybe add a vertical line in the plots to delineate the warmup phase)
- It seems to me that the proposed framework is close to a risk-seeking learning perspective where more weights are given to conforming examples. (The opposite, risk-sensitive learning, would give more weight to outliers so that the solution is good even on rarely seen examples). Here the authors hav a given prior on what conform means (that is the set of preferred clients). Could the authors comment on such a link and more generic formulation? It is generally odd to think that the proposed framework cannot be cast in any previous generic framework. The approach is surely new but it would help to consider its differences with other approaches in a common framework detailed mathematically.

---

> ### Author Response · Authors · 2024-07-09
> **Response to weaknesses raised by reviewer hCpB**
>
> We thank the reviewer for their detailed feedback and questions. We address all the issues raised by the reviewer and changes corresponding to these issues.
>
> **Weakness 1**
>
> We agree with the reviewer that finding common ground will help us compare and contrast different approaches and contexts in which one approach would be better than the other for client selection.  In response, we have revised our paper to include a comparative analysis that highlights the trade-offs between different methods.
>
>  For instance, we acknowledge the importance of gradient comparisons over loss comparisons. Gradient alignment is usually a better indicator of client alignment at a given time. Absolute loss-based methods, such as those proposed by Cho et al. (2020), are noted for their efficiency by eliminating the bottleneck of repeatedly comparing client with server losses. We emphasize that while our approach enhances communication efficiency, other methods may excel in scenarios where computational efficiency is important and communication is not the bottleneck. This discussion appears in the subsection on Client Selection (Section 2)
>
> To provide a clearer comparison, we have added a table (Table 1) summarizing the advantages and disadvantages of each method discussed. This addition aims to present a balanced view, acknowledging that different methods may be more suitable depending on the specific context.
>
> **Weakness 2**
>
> Thank you for your detailed feedback on the evaluation of sensitivities. We have conducted the requested experiments and included comprehensive discussions in the appendix, specifically in Appendix sections D1, D5, D6, and Figure 10.
>
> **Sensitivity to $\epsilon_T$​: (Appendix D1)**
>
> We have evaluated the sensitivity of our approach to the thresholds $\epsilon_t$​ and included a plot showing the final accuracy versus the choice of $\epsilon$. Our findings underscore the importance of selecting an optimal $\epsilon$ value to balance the trade-off between the inclusion of non-priority clients and alignment. Specifically, we observed that including too few clients fails to add significant computation, while including too many clients introduces a lot of bias, leading to degradation in performance. By fine-tuning $\epsilon$, we can strike the right balance.
>
> **Sensitivity to Priority Fraction: (Appendix D5)**
>
> We observed that the benefits of our approach can vary dramatically with the proportion of priority clients. We believe there are two competing factors influencing the model's performance. When the priority fraction is low, including more non-priority clients can increase accuracy due to the additional updates. However, this also introduces bias from the non-priority clients which reduces accuracy. Conversely, as we increase the number of priority clients, we achieve more aligned gradient updates but lose the benefit of extra updates from non-priority clients. This likely results in a complex dependency, causing sensitivity to vary significantly as the fraction of non-priority clients changes.
>
> **Sensitivity to Warm-Up Rounds: (Appendix D6)**
>
> We observed that initially, increasing the number of warm-up rounds improves performance, as it allows the model to better tune to the data from priority clients. However, as the number of warm-up rounds continues to increase, performance begins to decline. This is because the additional warm-up rounds come at the expense of updates from non-priority clients, leading to a trade-off between tuning to priority clients and incorporating diverse updates.
>
> **Non-Priority Client Alignment: (Figure 10)**
>
> We added a figure (Figure 10) summarizing the results we observed for synthetic data. As observed before, FedALIGN outperforms other baselines.

---

> ### Author Response · Authors · 2024-07-09
> **Reponse to additional comments raised by reviewer hCpB**
>
> **Additional comment 1**
>
> We apologize for proposing our own definition of fairness. Our intention was to rewrite existing notions of fairness in a way that they could be adapted more effectively for the PFL setting. To clarify this, we have added references to make it clear that our definition is inspired by established definitions of fairness.
>
> **Additional comment 2**
>
>   Fixed it - Thanks
>
> **Additional comment 3**
>
> You are correct in noting that the current algorithm also uses communications with all clients. We were trying to draw a distinction between downstream (from server to clients) and upstream (from clients to server) communication .In FedALIGN, the server broadcasts downstream to all clients, while only aligned non-priority clients communicate upstream. This differs from current methods, which typically require full communication from all clients. We have also removed the line saying “Therefore these algorithms do not serve as natural/direct baselines for our setting”, since as the reviewer rightly points out, finding a common ground baseline is beneficial to the paper. We apologize for any lack of clarity. We have updated the paragraph to make it clear in the paper.
>
> **Additional comment 4**
>
> We have added names to the input parameters to facilitate easier reading and understanding.
>
> Regarding the definition of $p$, we acknowledge the confusion. $p$ is intended to represent the total data fraction of all included non-priority clients along with priority clients in each communication round. While it is initialized at 1 (the normalized data fraction when no non-priority clients are included), it is then updated to reflect the combined data fraction of the priority clients and the selected non-priority clients. In the update step, we normalize the total data with both the priority clients and the selected non-priority clients. We apologize for any confusion caused by the initial explanation.
>
> **Additional comment 5 and 6**
>
> You are correct in suggesting that the concept of "compelling incentive" needs to be grounded in a more formal framework, such as utility functions from economic literature. In response, we have toned down our statements in the main text and moved those lacking formal evidence to the conclusion section.
>
> We intended to convey that the availability of well-performing, fine-tuned models can serve as a strong incentive for non-priority clients to contribute, as it allows for better personalization at each client. We have adjusted the wording to reflect this more clearly.
>
> **Additional comment 7**
>
> In PFL, we observed that due to the heterogeneity between priority and non-priority clients, and since we aim to learn models tailored to the objectives of the priority clients, the performance of FedALIGN could be enhanced by initially "warming up" on data from the priority clients. We acknowledge that a deeper analysis of this aspect is indeed valuable, as you pointed out.
> In response to a previous comment, we have conducted sensitivity experiments to explore this further and have included a detailed discussion of these findings in the revised manuscript..
>
> **Additional comment 8**
>
> Thank you for your insightful perspective! We agree that constructing a general framework to understand the relationship between FL and PFL can be quite beneficial. We propose the following framework:
>
> We can conceptualize FL and PFL as two ends of a spectrum. On one end, traditional FL involves weighing the objectives generated by both priority and non-priority clients according to their respective data fractions. This ensures that each client, regardless of priority status, contributes proportionally to the overall model training.
>
> On the other end of the spectrum, PFL focuses exclusively on the objectives generated by priority clients, effectively assigning a weight of zero to the non-priority clients. This approach prioritizes the specific needs and data characteristics of the priority clients, ensuring that the model is tailored to their unique requirements.
>
> An intermediate approach would involve assigning varying weights to the objectives generated by priority and non-priority clients, with a greater emphasis on the priority clients. By adjusting these weights, we can control the influence of each client group on the model training process.
>
> This may be a good direction to explore in the future. We added this to our conclusion.
>
> Thank you for your detailed feedback.  We have addressed all the main issues and additional remarks you highlighted.

---

### Author Response · Authors · 2024-07-09
**Main revisions to the manuscript**

We sincerely thank the reviewers for their extensive and insightful comments, which have significantly enhanced the quality of our paper. Below, we outline the major changes made to the paper in response to the reviews. All modifications are highlighted in blue in the revised manuscript. Additionally, we provide detailed responses to each reviewer's concerns.

**Comparison with Client Selection Strategies:**

We have expanded the comparison with various client selection strategies, focusing on the advantages and disadvantages of each method, including FedALIGN in prioritized federated learning. This section now provides a more comprehensive analysis, aiding in a clearer understanding of the comparative benefits of our proposed method. We also added a table (Table 1) with some key advantages of various client selection strategies compared to FedALIGN

**Sensitivity to $\epsilon_T$:**

To analyze the sensitivity to $\epsilon_T$, we have added a plot in the Appendix that plots accuracy as $\epsilon_T$ increases. We also provide a discussion on optimal $\epsilon_T$ selection strategies, offering insights into how different $\epsilon$ values impact the model's performance. This is available in Appendix D.1

**Effect of Warm-Up Rounds and fraction of Priority Clients:**

We have included an additional plot in the Appendix that plot accuracy concerning the number of warm-up rounds and the fraction of priority clients. These plots help to illustrate how variations in these parameters affect the performance, offering valuable guidance on parameter tuning. This is available in Appendix D.5 and D.6

**Computational Complexity of FedALIGN**

We add a dedicated section in the appendix (Appendix B) with full computational complexity analysis for our algorithm, including client-side and server-side complexity analysis.

**Test Accuracy and Client Alignment:**

We add a plot summarizing the accuracy as a function of client alignment for each method in the Figure 10 in the Appendix.
We trust that these changes address the reviewers' concerns and enhance the clarity of our findings.

---

### Decision · Action_Editor_XRje · 2024-08-17

**Recommendation:** Accept as is

**Comment:**

Initially, some of the concerns were comparison with related methods and sensitivity analysis of hyperparameters.

After reviewing the revised version, the reviewers are generally satisfied with the paper.   Two reviewers recommended "leaning to accept" and one recommended "accept".   Since the reviewers did not point out additional revisions, I recommend "accept as is".

**Audience:**

Individuals interested in federated learning with clients of difference importance could find the article interesting.

**Claims And Evidence:**

The authors consider the problem of Prioritized Federated Learning, where a subset of the clients are priority clients.   The objective is a weighted mean of the priority clients.  They would like to select a set of non-priority clients that are aligned with the priority clients to join the federation to improve the objective.   The FedALIGN algorithm matches the loss of non-priority clients on their own data with the loss of the global data.  The authors provide a theoretical Convergence analysis and results from empirical experiments on three datasets and two baseline methods.  The results indicate the benefits of the proposed algorithm.